

# Climate and music (Toward development of the interdisciplinary climate and cultural understanding education of ESD with special attention to the seasonal cycle and "seasonal feeling" around Japan and Europe)

Kuranoshin Kato[1], Haruko Kato[2], Rikako Akagi[1]

[1]Graduate School of Education, Okayama University, Okayama-city, 700-8530, Japan
[2]Faculty of Education, Gifu Shotoku Gakuen University, Gifu-city, 501 - 6194, Japan

*Correspondence to*: Kuranoshin Kato (kuranos@okayama-u.ac.jp)

**Abstract.**  Most music pieces have their own cultural background, and the origin and expression of songs are closely related
not only to their languages and customs but also to the regional characteristics of natural environment including the climate and its seasonal cycles. Thus, scientific study of the climate and seasonal cycle in a regional context would be also useful for understanding the context of music, as well as the other background. Inversely, such approach enables us to sympathize with the feeling of the people and to sing and appreciate the songs, even for the other regions.

By the way, climate education is an important part of the ESD (Education for Sustainable Development), relating to
education on environment, disaster prevention, climate variability, and also cultural understanding. Furthermore, it could contribute greatly to promoting the "Fundamental ESD Literacy" such as thinking of various complex relations, diversity, understanding of "Heterogeneous others", and so on.

Based on the above concepts, we have continued the interdisciplinary integration of the knowledge on climate and music, and have developed the crosscutting study plans on the climate and cultural understanding education.  A part of these results
have been just published in a Japanese book titled by "Climate and music (Cultural understanding and ESD spreading from the "Doors of Song")" (Kato, H. and K. Kato 2019), building mainly our papers written in Japanese, some of which were also introduced at EGU2014-2019. This article will re-integrate our above results, mainly for the climate and songs/traditional seasonal events around Germany, Northern Europe and Japan.

## 1 Introduction

Most music pieces have their own cultural background, and the origin and expression of songs are closely related not only to their languages and customs but also to the regional characteristics of natural environment including the climate and its seasonal cycles. Thus, as illustrated in Fig. 1, scientific study of the climate and seasonal cycle in a regional context would be also useful for understanding the context of music, as well as the other backgrounds. Inversely, to know how the climate,



season and people's feelings are expressed in songs, and so an, often gives us some interesting viewpoints for the data
analyses on climate and seasonal features including their variability.

Climate education is an important part of the Education for Sustainable Development (ESD) (UNESCO 2006, 2017), relating
to the education on environment, disaster prevention, climate variability, and also cultural understanding as a background of
the cultural generation. The climate and weather systems are generally characterized as the non-linear interaction systems
with complicated feedback mechanisms, multi-scale structure and variability. Thus the education on such climate or weather
systems and their variability could also contribute greatly to promoting the "Fundamental ESD Literacy" itself such as
thinking of various complex relations, diversity, and so on, among the various ESD targets.  This could also lead to promote
the students' ability of  "Understanding of Heterogeneous others", which is also an important literacy in ESD and Global
Citizenship Education (GCED) (UNESCO 2014, 2015).  In those activities, selection of the study areas or targets which are
not so familiar to the students in their usual lives sometimes gives the considerable advantage for promoting the students'
"Fundamental ESD Literacy" including the "Understanding of Heterogeneous others", by careful examination of the climate
data and deeper appreciation of the seasonal feeling expressed in the music works or traditional events as if they have been
there since before.

As for the climate systems, the seasonality is a common important feature characterizing the climate in the mid-latitude
regions.  However, many different factors relating to the seasonal cycles result in the great variety of the seasonal features
from region to region within the mid-latitudes.

For example, the significant rainy season call the Baiu in Japan (the Meiyu in China and the Changma in Korea) appears
just before the midsummer in East Asia and the heavy rainfall events frequently occur especially from Central China to the
western Japan in that season, greatly influenced by the global-scale Asian monsoon system (Kato 1989; Ninomiya 1989;
Nimomiya and Muraki, 1986; Ninomiya and Mizuno 1987, etc.). The precipitation attains the annual maximum in summer
also around Germany but the summertime precipitation there is not so large as in Japan (e.g., Kato and Kato 2019). On the
contrary, outbreak of rather cold air from the continent in association with the Asian monsoon system, over the warm
underlying sea, results in the extremely heavy snowfall climate in the Japan Sea side of the Japan Islands for its lower
latitude (e.g., Nimomiya 1968, 2006, 2007).

Furthermore, the global-scale Asian monsoon system can be regarded as consisting of the several subsystems with the
considerable phase difference of the seasonal progression among each other, at most by about three months (Murakami and
Matsumoto 1994; Kato et al. 2009, 2011; Kato and Kato 2014a). Since the climate systems in East Asia are greatly
controlled by the combined effects of these Asian monsoon subsystems, the seasonal cycle in East Asia shows the many
stages with rather different characteristics among each other, as illustrated in Fig. 2, and such short-step seasonal transition
would bring the remarkable change in the "seasonal feeling" from month to month (Kato and Kato 2014a, 2019; Yoshino
and Kai 1977).

First of all, the progression on the Baiu/Meiyu/Cangma in East Asia with the several abrupt seasonal transitions from spring
to midsummer is the typical example (Kato 1985, 1987, 1989; Kato and Kodama 1992; Hirasawa et al. 1995; Ninomiya and



Muraki 1986). In addition, another significant rainy season called the "Aiki-same" (秋雨) (consisting of the two Chinese characters which mean for "autumn" and "rainfall", respectively) appears in Japan, just after the midsummer (Matsumoto
65 1988).

It is also noted that an intermediate but rather distinctive season between autumn and midwinter can be identified as "early winter" around the Japan Islands, and that between midwinter and spring as "early spring". However, the seasonal progression from autumn to midwinter and that from midwinter to spring show rather asymmetric character as will be mentioned in Section 2 (Kato and Kato 2014a, 2019; Kato et al., 2013, 2014, 2015, 2017a). Thus, focusing on such
asymmetric seasonal progression around Japan as a topic for the interdisciplinary climate-cultural understanding education could give a useful approach for promoting the ESD literacy through the students perceiving the great differences between the two superficially resembling seasons, although learning on the heavy rainfall in the Baiu season, the heavy snowfall in midwinter, and so on, might be more important only for the disaster prevention education in Japan.

In Central and Northern Europe, the amplitude of the day-to-day variation of daily mean surface air temperature is rather
larger throughout a year like for example in Germany, in contrast to Japan, as will be mentioned in 3.1. Especially for winter, Kato et al. (2017b) and Hamaki et al. (2018) suggested that intermittent appearance of the extremely cold days in association with the large day-to-day variation seems to result in the "seasonal feeling" of "very severe winter" around Germany such as relating to the traditional event "Fasnacht" for driving winter away (Moser 1993; Nußbaumer 2010). As for the Northern Europe, the traditional event called "Juhannus", which means the "Summer Solstice Festival", is known as one of the most
famous ones. As Kato et al. (2019) indicated, while the mean air temperature from late June to July just after the summer solstice gets as high as in summer around Germany, the rather cold days begins to appear already in September, which seems to relate to the seasonal feeling of the short summer there (See 3.2).

Based on the above concepts, we have continued the interdisciplinary integration of the knowledge on climate and music, and have developed the crosscutting lesson studies on the climate and cultural understanding education for many years with
many co-researchers, mainly focusing on the seasonal cycle and seasonal feeling around Germany, northern Europe and Japan. A part of these integrated results have been published in the Japanese book titled by "Climate and music (Cultural understanding and ESD spreading from the "Doors of Song")" (Kato and Kato 2019) and "Climate and Music (Seasonal features and songs of spring in Japan and Germany)" (Kato and Kato 2014a) (Fig. 3). The words "Doors of Song" used for a part of the subtitle of Kato and Kato (2019) were got from the famous song "On wings of song" (its German title is "Auf
Flügeln des Gesanges") by Jacob Ludwig Felix Mendelssohn Bartholdy (Germany, 1809 - 1847).

As for Kato and Kato (2019), the content consists of the following three parts.

Part I: Education toward promoting the ESD Literacy through Climate and Music (Our concept).

Part II: Characteristic features of climate in their seasonal cycles and the "Seasonal feeling" in the songs and traditional events around Japan, Germany and Northern Europe at the view point of their inter-comparison.
Part III: Trials of interdisciplinary lesson studies by the collaboration between climate and music toward a new viewpoint on cultural understanding education.





In this review article, we will re-integrate the results of our joint activities mainly on the following topics (1) to (3), referring to our previous papers and books (although written in Japanese).

(1) Asymmetric seasonal march from autumn to the next spring around Japan.

(2) Winter climate around Germany in association with the traditional events "Fasnacht" for driving winter away.

(3) Seasonal cycle and "seasonal feeling" around Northern Europe with special attention to the climate and summer solstice festival there, with comparison of summer climate around Japan.

In the book by Kato and Kato (2019), the topic on the activity with focus on the seasonal march from spring to summer through the Baiu (significant rainy season) around Japan is also presented. As for this topic, the lesson practice was made for

the university students of music course in Faculty of Education, and the musical scores of the works composed by them together with climate description from spring to summer around Japan Islands were recorded in the book. However, that topic could not be shown here due to limitation of the space.

**2 Interdisciplinary lesson studies on climate and cultural understanding education with attention to the asymmetric seasonal march from autumn to the next spring around Japan**

**2.1 Asymmetric features of the seasonal march from autumn to the next spring and the relating "seasonal feeling" around Japan**

As shown in the right panel of Fig. 4, the mean air temperature in winter from northeast Siberia to Japan is rather lower than in the similar latitude regions and such cold air mass is called the Siberian Air Mass. At that time, so called the Siberian High and the Aluetian Low at the surface level tend to persist during winter and such pressure pattern is called the "winter

pressure pattern" in East Asia (Fig. 5). In that situation, the huge amounts of sensible heat and latent heat are supplied from the underlying sea along the path of the very cold air from the continent, and the convective mixed layer due to the shallow cumuli develops over the sea. These result in the climatologically huge snowfall in the Japan Sea side of the Japan Islands in winter (e.g., Ninomiya 1968, 2006, 2007).

Besides, it is also interesting that the lowest mean temperature stage around Japan (e.g., at Tokyo) lags to that in northeast

Siberia (e.g., at Ojmjakon) by about a month (the left panel of Fig. 4), greatly reflected by the more phase lag of the seasonal progression in the subtropical or tropical western Pacific.  Reflected by the earlier seasonal rapid temperature fall around the northeast Siberia, the appearance frequency of the "winter pressure pattern" begins to increase already in November (early winter) (Fig. 6).  However, the air temperature around the Japan Islands is rather higher than in early March (early spring) when the appearance frequency of the "winter pressure pattern" is nearly the same as in November. Thus the precipitation in

the plain area of the Japan Sea side in Japan in the "winter pressure pattern" is brought not as snow but mainly as rain. The intermittent rainfall due to the shallow cumuli in such situation is called "Shi-gu-re" (時雨) in Japanese (consisting of the two Chinese characters which mean for "sometimes" (or intermittent) and "rainfall", respectively) and is often used for expressing the "seasonal feeling" in early winter in the Japanese classic literature. Especially we can see them in the



Japanese classic poems called "Wa-ka" (和歌), a kind of Japanese classical poems consisting of the 31 syllables (e.g., Table
1) (Kato et al. 2011, 2012, 2018).

However, the solar radiation is rather stronger in early spring than in early winter, as indicated by the seasonal variation of daily mean solar radiation reaching the outer atmosphere, daytime length and the duration when the sun altitude angle is higher than 45° at 35N (Fig. 7) (Kato et al. 2019). Such asymmetric seasonal progression from autumn to the next spring around Japan would result in rather different "seasonal feeling" between early winter and early spring. For example, while
the "Shi-gu-re" is often used for expression of the "seasonal feeling" in early winter in "Wa-ka" as mentioned above, the light snow under the relatively strong sunshine, and so on, is one of the popular scenes presented in the "Wa-Ka" for early spring (e.g., Fig. 8). In turn, such difference of the "seasonal feelings" between early winter and early spring might be useful for deeper understanding of the seasonal cycle of the climate system around Japan. As such, the next subsections will report our interdisciplinary lesson practice mainly for the university students in teacher training course, with focus on the
asymmetric seasonal progression from autumn to the next spring around the Japan Islands referring to Kato et al. (2013, 2014, 2017b), Kato and Kato (2019), and so on.

**2.2 A report of interdisciplinary lesson practice for the university students in teacher training course**

**2.2.1 Outline of the lesson practice**

The lesson practice reported by Kato et al. (2014) was performed as follows on the last day of an interdisciplinary class
"Human Lives and Environments" at Faculty of Education, Okayama University during 28～30 August 2013.

Instructor: Kato, K. (Meteorology), Akagi, R. (Art) and Kato, H. (Music)

Date and Time: 30 August 2013 (08:40～17:45)

Activities:

(1) Brief summary on the climatological features in that season and the appreciation of the "seasonal feeling" expressed in
the Japanese classic art works

(2) Expression of the seasonal feeling with coloured papers (Art)

(3) Expression of the seasonal feeling with small percussion instruments (Music)

(4) Expression with the colored papers on the bases of the activities 2 and 3 (Both).

About 35 students took part in this class. About the four fifth of them are not specialized in natural science. This class is
open for the all grade student. In this year, the numbers of the second and third grade students in this class were 9 and 20, respectively, and there were also several participants of the first and fourth grade, respectively. Prior to this joint activity, Kato, K. had explained to the students the climate around East Asia including the detailed seasonal cycles such as the asymmetric seasonal progression from autumn to the next spring around Japan.

In Activity (2), the students tried the expression of the seasonal feelings in early winter and early spring, respectively, by
using the 6 colors selected from the 96 colored papers, based on the Johannes Itten's (1888-1967) exercise for the expression



of the four seasons (Itten 1961). At the beginning of this students' activity, the following announcement was made. 1) Represent your sense of the seasons with 20squares (4 X 5) of 2 papers (two stages for the early winter and early spring). 2) Select the two stages for the detailed comparison as above. 3) What colors represent seasons? Select 6 colors from a set of 93 colored papers. 4) Think about which color combination is the best. Arrange and paste colored papers on a sheet. 5) Post

up your work on the wall of this lecture room at the proper position in the order of seasonal progression from autumn to spring through winter.

All works by the students in this activity exhibited in such manner are presented in Fig. 9.

In Activity (3), the students tried to express their own seasonal feeling on early winter and early spring with attention to the difference between the stages just before and after the midwinter, by using the various small percussion instruments (Fig. 10).

Their composition works were described in the form of the "graphic notation" proposed by Morton Feldman. Each student composed two works expressing early winter and early spring, respectively. Then, students were divided into several groups and each group played a selected set of their works. It took about 30 seconds to play a work. In the Activity (4), the students expressed the seasonal feeling of early winter or early spring with the less restricted usage of the colored papers than in Activity (2). The concept, outline and results of this lesson practice were briefly introduced also at the EGU2014 (EGU2014-

3708) and 2015 (EGU2015-2667).

### 2.2.2 Difference of the seasonal feeling between early winter and early spring around Japan expressed in the students' works in Activities (2) and (3)

 Analysis results of the students' works in Activities (2) and (3) (mainly for Activity (3)) will be briefly introduced after Kato et al. (2014). As for the works in Activity (2) (see Fig. 9), the cool colors were mainly used with accentuate use of

warm colors such as dark red and orange expressing fallen leaves and dead leaves in the works for early winter. It is also noted that the students tend to prefer the dark colors as for the cool ones. On the other hand, diverse combination of the rather different colors among the students' works tended to be used for expressing the seasonal feeling of early spring, i.e., not only the contrast between cool and warm colors, but also that on saturation and lightness. The cool colors seem to stand for the cold air and wind, early spring snow, and so on, while the warm colors remind us of the rather strong solar radiation

or flowers of plum and cherry. Thus, the students' works as the whole with use of the colored papers seem to express the rather different climate features between early winter and early spring as mentioned in 2.1.

As for the students' music works in Activity (3), according to Kato et al. (2014), the students' works consisted of both the descriptive expression of the meteorological or climatoligical phenomena peculiar to the target seasons and the abstract one such as the marching season and the related emotion.  Besides, various kinds of fluctuations such as the "Shi-gu-re"

(intermittent rainfall in the "winter pressure pattern") in early winter, day-to-day alternation of the cold and warm day, were also presented in their works.  An example of the students' music works comparing the seasonal feeling between early winter and early spring is shown in Fig. 11. Some additional explanation by the present authors is also made. The students' works succeeded to present what they intended to present about the asymmetry between early winter and the early spring.





Furthermore, the students' works seem to express not only the seasonal feeling derived from the mean seasonal state, but
also from a specified situations in the day-to-day variations or repetition of the day-to-day cycles embedded in the seasonal
progression. For example, as illustrated in Fig. 12 (Kato et al. 2014), although the seasonal mean temperature is higher in
early winter than in early spring around Japan Islands, the "maximum value" as the day-to-day variation of the daily mean
temperature in early spring can exceed the "minimum one" in early winter.

If the people tend to have a stronger impression of the colder days of the early winter due to the seasonality toward winter
and that of warmer days in early spring, it might not be unnatural that the people would feel early spring warmer than in
early winter, not only due to the condition on the stronger solar radiation in early spring illustrated in Fig. 7. In general, the
people's feelings are rather diverse even in the similar environment. This would be greatly due to the emotional factors. But
it should be also kept in mind that we could have many choices what kind of daily meteorological conditions we have
stronger impression of, because of their rather great day-to-day variability even within a specified season. Thus, depending
upon the other personal situations, a part of the natural environment characteristics including their variability could emerge
selectively (just we can say the "sensitivity filter") in the people's mind to form their diverse seasonal feeling. The present
activity also suggests that not only the seasonal mean meteorological factors but also the characteristic of their day-to-day
variability could affect greatly the peoples' seasonal feeling. Inversely, such "sensitivity filter" on the seasonal feeling seems
to give us an interesting viewpoint for deeper understanding of the climate environment including the seasonal cycles with
large day-to-day variability.

By the way, our group has made a lesson practice on the similar theme (only the collaboration between climate and music)
also at a senior high school (Kato et al. 2017a) for about 2 hours. Although the detailed introduction of this lesson will not be
shown here due to the space limitation, the students tried to arrange the 6 Japanese school songs in the detailed seasonal
order from autumn to winter and the other 6 ones from winter to spring by appreciating the seasonal feelings expressed in
these songs, together with the study on the related climatological features (Fig. 13).

### 3 Interdisciplinary lesson studies with attention to the seasonal cycle and the "seasonal feeling" around Germany and Northern Europe

#### 3.1 Remarks on the seasonal cycle and the "seasonal feeling" around Germany

Around Germany located near the western edge of the Eurasian Continent, there are many music and literature works in
which "the May" or spring is treated as the special season (Kato and Kato (2011, 2014a) also introduced the many German
songs on spring, May). The German lied "Im wunderschönen Monat Mai" ("In the especially beautiful month of May") by
Robert Alexander Schumann (1810 - 1856) is a typical example. However, the feeling of welcoming the special spring
season expressed in these works seems more than just waiting for winter to leave. In fact, there is a traditional event called
"Fasnacht" for driving winter away held around February or March there (Moser 1993; Nußbaumer 2010). The "Fasnacht"
would be relating to the "seasonal feeling" of "very severe winter" around Germany, as mentioned in the next paragraph.





Furthermore, the Japanese researchers on German Literature suggested that there are basically two seasons, i.e., "winter" and "summer", in a year around Germany, with short transition stages called "spring" and "autumn". The seasonal cycle around Germany could be characterized by the feeling that the winter and the summer battle with each other and the summer wins to drive the winter away (Oshio 1982; Miyashita 1982; Takeda 1980).

It is interesting that not only the seasonal mean surface air temperature around Germany is lower than in the Japan Islands area except for the northern part (〜to the south of about 40˚ N), but also the day-to-day variation of daily mean temperature there is rather larger throughout a year, especially in winter (Figs. 14 and 15). Kato et al. (2017b) and Hamaki et al. (2018) suggested that intermittent appearance of the extremely cold days (around -5〜-15℃) in winter in association with the large day-to-day variation influenced by the intraseasonal variation of the Icelandic low seems to result in the

"seasonal feeling" of "very severe winter" around Germany that they do wish to drive the winter away. The area where such extremely cold days appear in winter in association with the large day-to-day variation extends from Germany to the Northern Europe (Fig. 16). Besides, while the seasonal mean air temperature shows the highest from June to August there, its day-to-day variation (mainly relating to the intraseasonal one) in summer is still rather larger than in the Japan Islands (Fig. 15). Thus, the summer around Germany seems to be characterized by the seasonal maximum temperature stage with

the intermittent appearance of rather cooler spell (Fig. 17). And then, the "May" could be regarded as the very season just when the winter has been completely driven away and the "summer" has just begun.

Also in Japan, a traditional ceremony "Setsu-Bun" (節分), for calling spring, purging noxious vapors relating to winter such as evil spirits called "oni" (鬼) and welcoming good fortune, is held around the beginning of February. Some photographs of the scene of "Setsu-Bun" festival took by one of the present authors are illustrated in Fig. 18. However, differently from

"Fasnacht" around Germany, "Setsu-Bun" is held just when the temperature decrease stops to turn to the rising stage as the seasonal progression (i.e., the seasonally coldest period), but the daily mean solar radiation increases considerably from February to March there (Figs. 7 and 15). Furthermore, although the cold days as in midwinter still tend to appear frequently in February, rather warm days also tend to appear intermittently after around the "Setsu-Bun".

### 3.2 Remarks on the seasonal cycle and the "seasonal feeling" around Northern Europe

In Northern Europe, the traditional event called "Juhannus", which means the "Summer Solstice Festival", is held literally around the summer solstice night for cerebrating the summer coming. In "Juhannus" they put up a white birch as a pole, decorate their house with green leaves and build a bonfire called "kokko". Furthermore, children have a lot of fun in the forest, with making a cabin called the " Murska", making fairy dolls, and so on.

The seasonal feeling on the summer solstice is also expressed, for example, in a Finnish folk song "The night of the summer
solstice". The text of this song begins with "The long-awaited night of the summer solstice has come." It is sung that they have done very much things for preparing the festival. As such, they are only expecting the very moment as in this song "It



is finished to get ready for bonfire, so it is only to wait getting dark". They are going to sing and dance throughout night together.

As Kato et al. (2019) indicated, while more extremely cold days than around Germany tend to appear intermittently in winter around Northern Europe accompanied by the large day-to-day variations, the mean air temperature from late June to July just after the summer solstice gets almost as high as in summer around Germany (Figs. 14 and 15). Besides, as indicated in Fig. 7, the daytime length attains more than 18 hours even in the southern part of "Northern Europe" (60°N) (that is, the very short night), together with the relatively strong sunshine in the daytime. However, the rather cold days begin to appear already in September (Fig. 15), which seems to relate to the seasonal feeling of the rather short summer compared even with that around Germany.

As such, we should note the importance to refer to the climatological magnitude of variability even for understanding the "mean features" of the climate systems or their related "seasonal feeling", as illustrated in 3.1 and 3.2.

**3.3 A report of interdisciplinary lesson practice on the seasonal cycle and the "seasonal feeling" around Germany for the university students in teacher training course**

**3.3.1 Outline of the lesson practice**

The lesson practice reported by Kato et al. (2017) is briefly summarized in this section. The outline of the class is as follows. This joint activity was performed on the last day of an interdisciplinary class "Human Lives and Environments" at Faculty of Education, Okayama University, during 23～26 August 2016.

Instructor: Kato, K. (Meteorology), Akagi, A. (Art) and Kato, H. (Music)

Date and Time: 26 August 2016 (08:40～18:30)

Activities:

(1) Brief summary on the climatological features in winter in Europe, comparing with those in the Japan Islands and appreciation of the "seasonal feeling" expressed in the European painting.

(2) Appreciation of the music works on the European winter and video watching of "Fasnacht". Then, creation of the music works on their own feeling of "Fasnacht" with small percussion instruments, and playing their music works as in 2.2.1.

(3) Expression with the colored papers comparing the "seasonal feeling" between midwinter and early spring in the Japan Islands as in 2.2.1, and then, brief explanation of the related climatological features.

About 30 students took part in this class. About the two third of them are not specialized in natural science. This class is open for the all grade student. In this year, the numbers of the first and third grade students in this class were 16 and 7, respectively, and there were also several participants of the second and fourth grade, respectively. In Activity (2), their music works were described in the form of the "graphic notation" proposed by Morton Feldman, as in 2.2.1.



### 3.3.2 Discussion from the students' composition works in Activity (2)

Figure 19 indicates four examples of the students' composition works (Ex. 1 to Ex.4) on the original "Fasnacht" in Activity (2), after Kato et al. (2017). The titles and explanations of the works by their own are as follows (translated from Japanese
into English by the present authors).

(Ex. 1) "Sending off the winter"
I like the feeling of winter send-off (farewell to winter) rather than "driving out".  It is rather cold in winter in my hometown but I feel some beautiful atmosphere.  "See you again, winter!"
(Ex. 2) "Invite the spring with melting the snow"
At the first part, I expressed the continuously falling in the dark and gloomy winter.  I will express the scene that the snow stored in the sky has been completely felled down and sun welcomes the spring by melting that snow.  The final part presents the coming spring lively.
(Ex. 3) "Driving the severe winter away"
The lightly falling snow was expressed with bell and chajchas at first.  As the phrase is repeated again and again, the volume of the bell is enlarged together with the short-term fluctuation of that volume.  Near the final phrase, large spring drum sound, etc. is added, which presents the just driving the severe winter away.
(Ex. 4) "Threaten the winter to retreat"
Feeling of threatening the winter to retreat was expressed.    Loud clapping and bell sounds at the beginning stand for
defeating the winter. The feeling that the winter has gone out was expressed by the cymbal sound at the end of this piece. The emotion of the spring coming was also presented by the gradually increasing triangle sound toward the end of this piece, due to its bright image.

From these works, we can find out the following. In Ex.1, the student's unique sense on sending (not driving) the winter
seems to be expressed well. In Ex. 2, the darkness of winter and the merry spring atmosphere are illustrated contrastingly. In Ex.3, the gradual seasonal progression of the retreating winter is presented with the combination of changing loudness and tone of the sounds. In Ex. 4, the cymbal is effectively used for the symbol that the winter has retreated out. In these activities, students firstly tried to understand or imagine why the people living there can't help driving the winter away, based on the climate data. They seem to have an opportunity for realizing the climate in foreign regions and the "seasonal feeling" there,
through the composition and performance of the music works in such processes.



**3.4 A report of interdisciplinary lesson practice on the seasonal cycle and the "seasonal feeling" around Northern Europe for the university students in teacher training course**

**3.4.1 Outline of the lesson practice**

The lesson practice reported by Kato et al. (2019) is briefly summarized in this section. The outline of the class is as follows.

This joint activity was performed on the last day of an interdisciplinary class "Human Lives and Environments" for Faculty of Education students, 28～31 August 2017.

Instructor: Kato, K. (Meteorology), Akagi, A. (Art) and Kato, H. (Music)

Date and Time: 31 August 2017 (08:40～18:30)

Activities:

(1) Brief summary on the climatological features around winter in Europe comparing with those in Japan, and appreciation of the "seasonal feeling" expressed in Northern European pictures

(2) Watching video material of the traditional seasonal events and life in Northern Europe, appreciation of the traditional songs there, and then, students' composition and performance of the music works with small percussion instruments on the summer solstice there. Performance was made as in 2.2.1.

(3) Expression with the colored papers on the summer solstice in northern Europe after a brief explanation on the Johannes Itten's (1888-1967) exercise for presenting the four seasons, as in 2.2.1.

13 students took part in this class. 2 students are specialized in natural science. This class is open for the all grade student. In that year, the numbers of the first and third grade students in this class were 4 and 6, respectively, and there were also 3 participants of the other grades, respectively. In Activity (2), their music works were described in the form of the "graphic

notation" proposed by Morton Feldman, as in 2.2.1. As for the appreciation of the seasonal feeling expressed in the music works, the following Finnish folk songs were used, i.e., "February has come", "An old frost-man", "A tune of spring", "The night of the summer solstice" and "A summer day of Kangasalla". The concept, outline and results of this lesson practice were briefly introduced also at the EGU2018 (EGU2018-2822) and 2019 (EGU2019-13141).

**3.4.2 Discussion from the students' composition works in Activity (2)**

Students' viewpoints in expressing their music works (13 works in total) were summarized as the descriptive expression of the image of a scene (10 works) and expression of inner or more creative story (not simple description of a scene) (3 works in total). The former focused on scene of "Juhannus" (Summer solstice festival), its liveliness, summer solstice, scene of the nature in summer, children's activity in summer, and so an. The examples of the titles of such works are "Dance in Juhannus", " Adventure of summer forest", and so on.

On the other hand, the latter expressed characteristics of the season, view of life and fantastic story. The titles of the works in such category are "Long day and very short night", "Circulation (Seasonal cycle and life cycle)" and "Forest in summer night".





Figure 20 indicates three examples of the students' composition works (Ex. 1 to Ex.3) on "Juhannus" in Activity (2), after Kato et al. (2019). The titles and explanations of the works by their own are as follows (translated from Japanese into

English by the present authors).

(Ex. 1) "Long day and night of the moment"

After the long and lively day is over, the lonely and dark night stays only for an instance.  I noticed that the daytime in Finland is much longer than in Japan.  Liveliness in the daytime is expressed by the sounds of children and animals, while

Loneliness at night by Mokokku (like sound of flogs) and mysterious singing ball.

(Ex. 2) "Dance in Juhannus"

Image of a delightful summer solstice day is presented.  The evening wood fire is expressed at the beginning of this music with rainstick. Wood block presents the light footsteps of dance and the festival of getting blast by chajchas and bell. In the latter-half of the work, the low cowbell's tone sounds as the night goes on. And when morning comes with chirping of birds

(triangle), the festival comes to end with the sound disappearing gradually.

(Ex. 3) "Circulation (Seasonal cycle and life cycle) "

As often said "winter brings the change and gives the emotion of waiting for spring", people in Finland strongly feel the seasonal cycle in their usual life. Furthermore, people engrave their own name of the dead on the trees in forest and return to the nature as a new life cycle.  Such transmigration of the soul, i.e., the endless cycle of the death and rebirth and that of all

lives in forest is presented.

Also in this activity, the students seem to have an opportunity for realizing the climate in foreign regions and the "seasonal feeling" there, as in 3.3. We should also note the importance of referring to the climatological variability even for understanding the mean features of the climate systems and their related "seasonal feeling", through the activity on the

climate and seasonal feeling for Germany and northern Europe.

**4 Summary and discussions**

 In this review article, our interdisciplinary research on climate and music, and the interdisciplinary lesson studies on climate and cultural understanding education for promoting the fundamental ESD literacy were introduced, mainly for the climate and songs including the traditional seasonal events around Germany, Northern Europe and Japan as follows.

(1) Asymmetric seasonal progression from autumn to the next spring around Japan.

(2) Winter climate around Germany in association with the traditional events "Fasnacht" for driving winter away.

(3) Seasonal cycle and "seasonal feeling" around Northern Europe with special attention to the climate and summer solstice festival there, with comparison of the summer climate around Japan.



The choice of the study areas or targets including the European regions not so familiar to the Japanese students in their usual lives was aimed to give the advantage for developing the students' ability of deeper understanding and imagination on what they have not experienced, leading to promote the students' ability of "Understanding of the Heterogeneous others". Thus, in the analyses of our lesson practice results based on the students' works, we checked how deeply the students considered what they try to express with integrating what they have learned throughout this class, rather than how well their works have

been completed.

Main climatological features and the results of the interdisciplinary activities on the topics (1) to (3) are as follows.

The topic (1) indicates that while the appearance frequency of the "winter pressure pattern" in early winter and in early spring around Japan is nearly the same, air temperature is rather lower in early spring. But the solar radiation is rather

stronger in early spring. As for the topics (2) and (3), the "seasonal feeling" of the "severe winter" around Germany and Northern Europe seems to be associated with the intermittent appearance of the extremely cold days as the very large day-to-day variations, and the seasonal feeling of short summer around Northern Europe would be due to the early seasonal appearance of the rather cold days already in September also as the large day-to-day variability.

The results of the lesson practices on those topics illustrate that not only the seasonal mean temperature or solar radiation

condition but also the large day-to-day variability including the appearance of the extremely low temperature events, and so on, was strongly reflected in the students' works, although the statistical evaluation on the students' understanding would be also needed in the future.

Finally, some viewpoints of understanding of the characteristics of the climate relating to the seasonal feeling for our

interdisciplinary lesson study will be briefly discussed.

One is on the variety of patterns of the mean seasonal cycles. Most simply, seasonal cycles of the meteorological elements are generally characterized as sinusoidal waives. However, in detail, they are sometimes rather distorted from the sinusoidal shape from region to region, as well as the amplitude and phase. For example, as illustrated in 3.1 and 3.2, the period of the highest stage of the seasonal mean temperature around Germany is rather longer (from the beginning of June to late August)

than in Northern Europe (from late June to late July), although the day-to-day temperature variation is also large in both regions. As for the Japan Islands, the highest temperature season is from late July to late August, although such description was not made in this article. In addition, some meteorological or climatologocal elements in the same region show sometimes rather large phase lag of their seasonal variations among each other, as mentioned on the above topic (1) around the Japan Islands, i.e., those among the appearance frequency of the "winter pressure pattern", mean temperature, sunshine

conditions, and so on, relating to the feeling on the asymmetric seasonal progression from autumn to the next spring.

Another important viewpoint is that we should pay attention also to the day-to-day variation features. This seems to be relating greatly to the students' recognition how the winter in German is severe compared with that in Japan, for example, on the above topic (2), together with that on the short summer in Northern Europe on the topic (3).



As such, our series of studies introduced in this article seem to illustrate the usefulness of taking notice of the difference of
the seasonal cycle patterns and magnitude of the day-to-day variability relating to the seasonal feeling for the interdisciplinary climate-cultural understanding education. However, there are many other types of the seasonal cycles in association with the regional variety of seasonal feelings even within the mid-latitude or higher latitude regions. For example, the German seems to have another special seasonal feeling of autumn to early winter. The German poems "Im Herbst" (In the autumn) by Rainer Maria Rilke (1885 - 1926), "Im Nebel" (In the fog) by Hermann Karl Hesse (1877 - 1962), and so an,
might be the typical examples. But the interdisciplinary lesson studies based on the further inter-comparison of the seasonal cycles and seasonal feelings including those topics are the interesting remaining problems in the future.

**Acknowledgement**

The present study was partly supported by JSPS Grants-in-Aid for Scientific Researches (C) (No. 17K04817) and (B) (No. 17H02700).

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



**Table 1: Number of the Japanese classical poems "Wa-ka"s in which the "Shi-gu-re" is expressed in the first 8 Waka Collections compiled by Imperial command ("Chokusen Wakashu") edited from 10th to 13th centuries (Kato et al. 2018).**

| | 勅撰和歌集(Name of the book in which the Japanese classical poems ("Wa-ka") are collected) | Number of the "Wa-ka" in which the "Shi-gu-re" is used for expression | Total number of the "Wa-ka" for the volume on autumn and winter | Ratio(%) |
|---|---|---|---|---|
| 1 | 古今 ("Kokin") | 4 | 174 | 2.3 |
| 2 | 後撰 ("Gosen") | 17 | 290 | 5.9 |
| 3 | 拾遺 ("Shuui") | 6 | 126 | 4.8 |
| 4 | 後拾遺("Go-shuui") | 6 | 190 | 3.2 |
| 5 | 金葉(第三奏本)("Kin-yo") | 5 | 165 | 3.0 |
| 6 | 詞歌 ("Shika") | 5 | 79 | 6.3 |
| 7 | 千載 ("Senzai") | 21 | 248 | 8.5 |
| 8 | 新古今 ("Shin-kokin") | 35 | 422 | 8.3 |















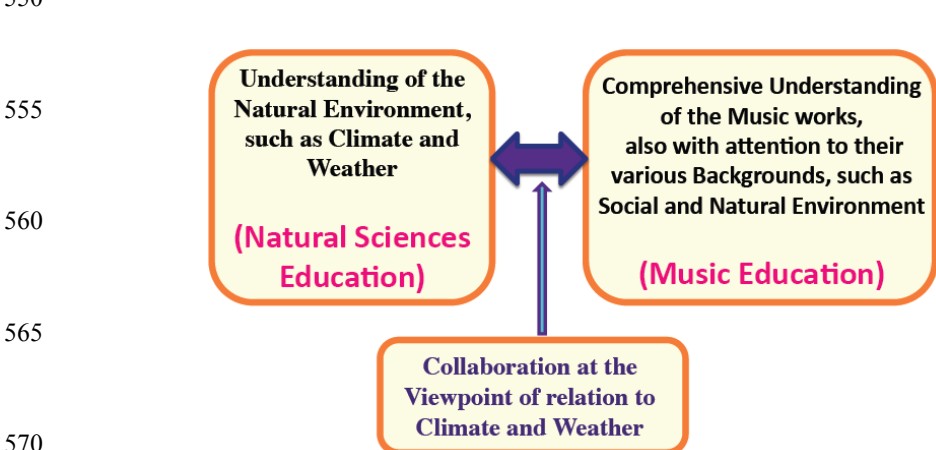

**Figure 1: Collaboration between climate and music, toward interdisciplinary cultural understanding education.**




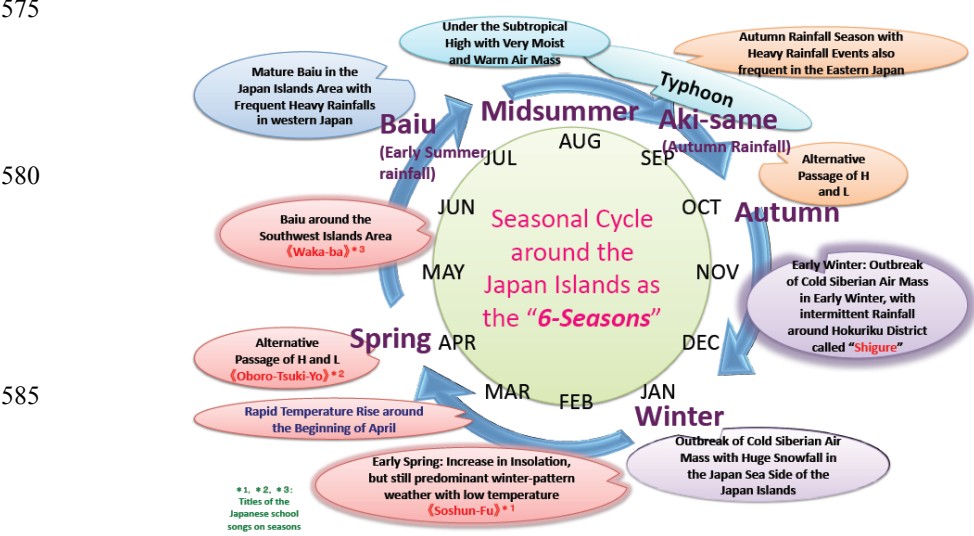

**Figure 2: Schematic figure of the seasonal cycle around the Japan Islands, in East Asia.**





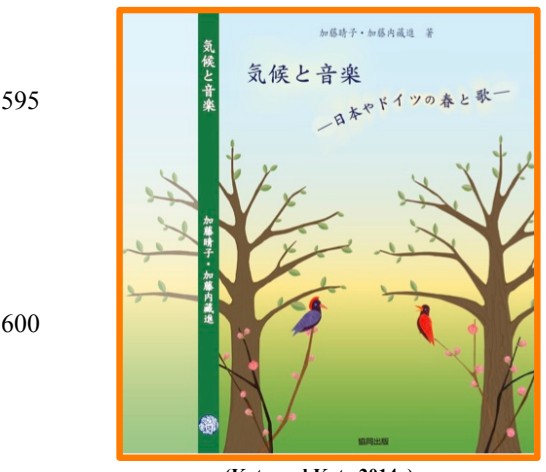
(Kato and Kato 2014a)

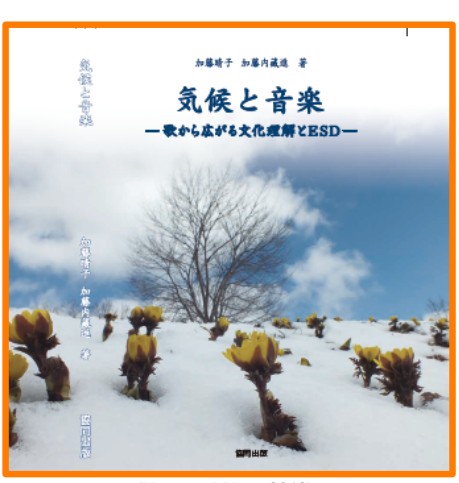
(Kato and Kato 2019)



Figure 3: The surface covers of the two books which summarized a part of our interdisciplinary studies written in Japanese, titled by "Climate and Music (Seasonal Features and Songs of Spring in Japan and Germany)" (the left figure) (Kato and Kato 2014a) and "Climate and music (Cultural understanding and ESD spreading from the "Doors of Song")" (Kato and Kato 2019) (the right one), respectively.


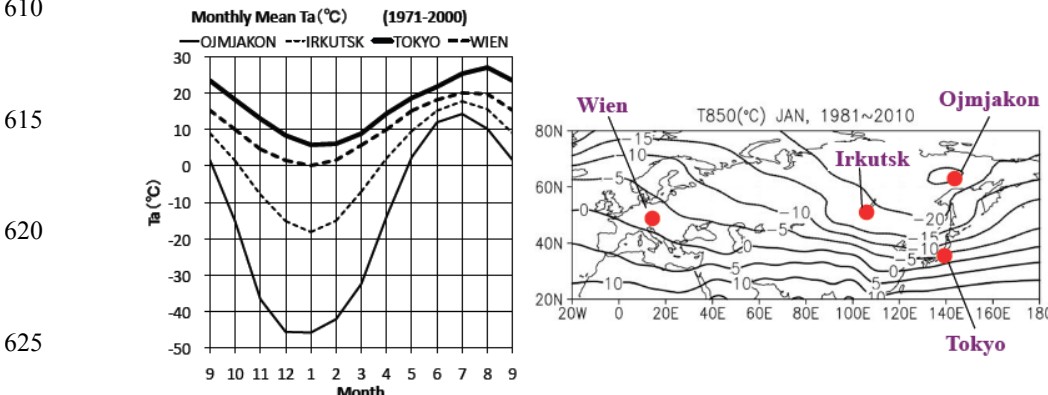




Figure 4: (Right panel) Monthly mean air temperature at 850hPa level (℃) in January averaged for 1981-2010 (after Kuwana et al. 2016), based on the NCEP/NCAR re-analysis data (Kalnay et al. 1996). (Left panel) Seasonal variations of the monthly mean surface air temperature at Ojmjakon, Irukutsk, Tokyo and Wien. Locations of these stations are also shown in the right figure.



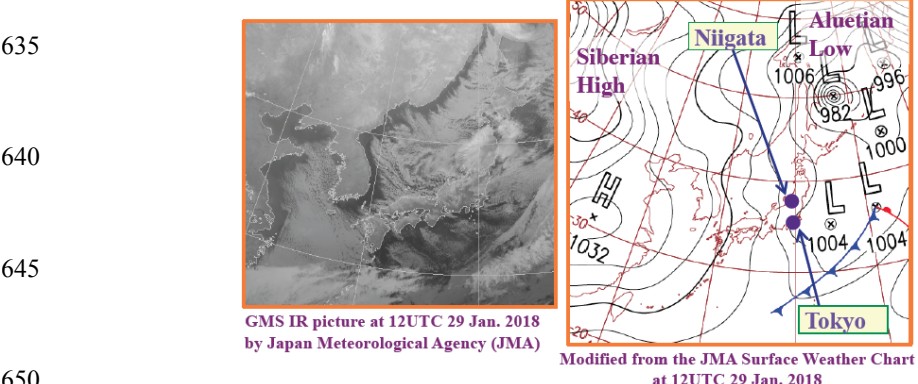

**Figure 5: A typical example of the surface weather chart (right) and that of the infrared picture of the Geostationary Meteorological Satellite (GMS) (left) in the "winter pressure pattern" situations around East Asia in midwinter. These figures are cut out from the original figures provided by the Japan Meteorological Agency (JMA). Locations of the surface meteorological stations Niigata (the Japan Sea side) and Tokyo (the Pacific side) are also added on the weather chart by the authors.**

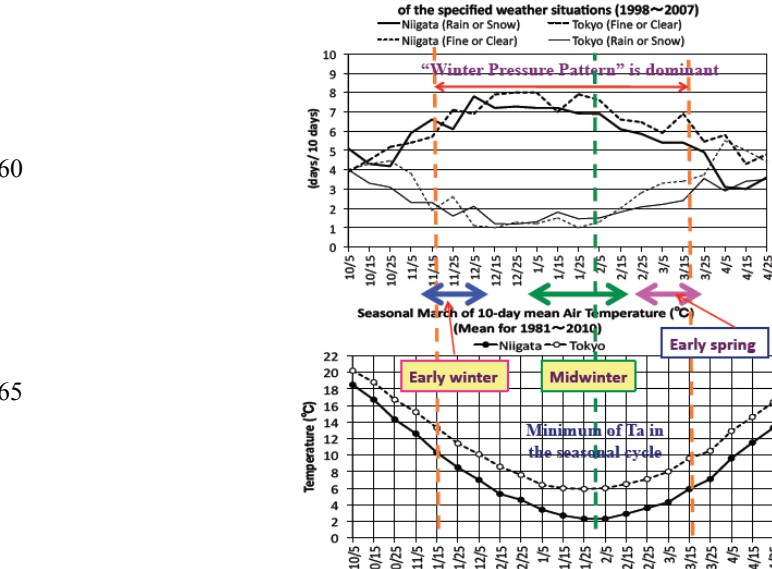

**Figure 6: (Upper panel) Seasonal change in the number of days (per 10 days) of the specified weather situations at Niigata and Tokyo (1998〜2007). (Lower panel) Time series of 10-day mean air temperature (℃) at Niigata and Tokyo (Mean for 1981〜2010). These figures are after Kato et al. (2013) with some modification. The locations of these stations are referred to Fig. 5.**



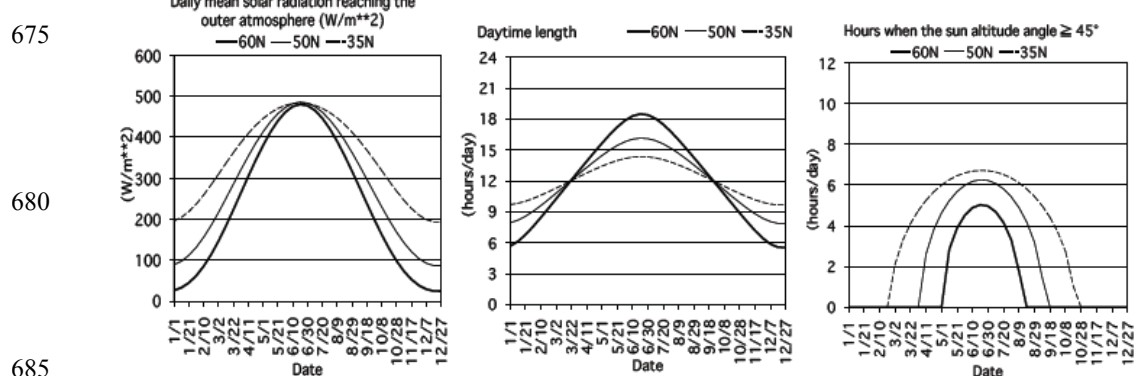

Figure 7: Seasonal variations of daily mean solar radiation reaching the outer atmosphere (Wm⁻², left panel), daytime length (hours/day, central panel) and the duration when the sun altitude angle is higher than 45° (hours/day, right panel) at 35°N (broken line, around the southern part of the Japan Islands), 50°N (thin solid line, around Germany) and 60°N (thick solid line, around the southern part of Finland) (Kato et al. 2019). The data at 50°N and 60°N will be referred to in Section 3.

【Early Winter】

神な月 降りみ降らずみ さだめなき
時雨ぞ冬の はじめなりける
(pronunciation) Kan-na-Zuki Furi-mi Furazu-mi Sadame-naki/ Shigure-zo Fuyu-no Hjime Nari-keru
Poet: unknown(in "Gosen")

(Meteorological situation) The "Shigure" precipitation in the early winter monsoon situation is characterized by the intermittent one, i.e., once it begins to rain, it stops soon. Such situation continues for several days.

【Early Spring】

春日野の下萌えわたる草のうへに
つれなく見ゆる春のあわ雪
(pronunciation) Kasuga-No-no Shita-Moe-Wataru Kusa-no-Ue-ni/ Tsurenaku-Miyuru Haru-no-Awa-Yuki
by Gon-Chuunagon-Kuninobu
権中納言國信（in "Shin-Kokin"）

(Meteorological situation) Although the light snow cover remains in the wide grass field, the new grasses are just growing there in the spring bright sunshine. The contrast among the light green, bright sunshine and the white snow seems to be impressive.

Figure 8: An example of the "Wa-ka" expressing the seasonal feeling in early winter (e.g., "Shi-gu-re") and that in early spring (e.g., light snow under the relatively strong sunshine). The former is recorded as 445th work in the 2nd Waka Collection "Gosen" and the latter as 10th in the 8th Collection "Shin-Kokin" compiled by Imperial command (also see Table 1). Pronunciation and the meteorological situation for each "Wa-ka" are also shown.



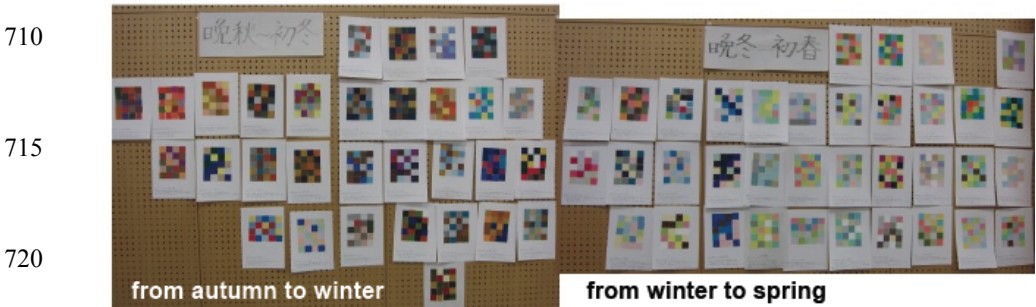


**Figure 9: All works by the students in Activity (2) (32 students x 2 seasons = 64 sheets in total) are ordered along the seasonal**
**progression, after Kato et al. (2014).**

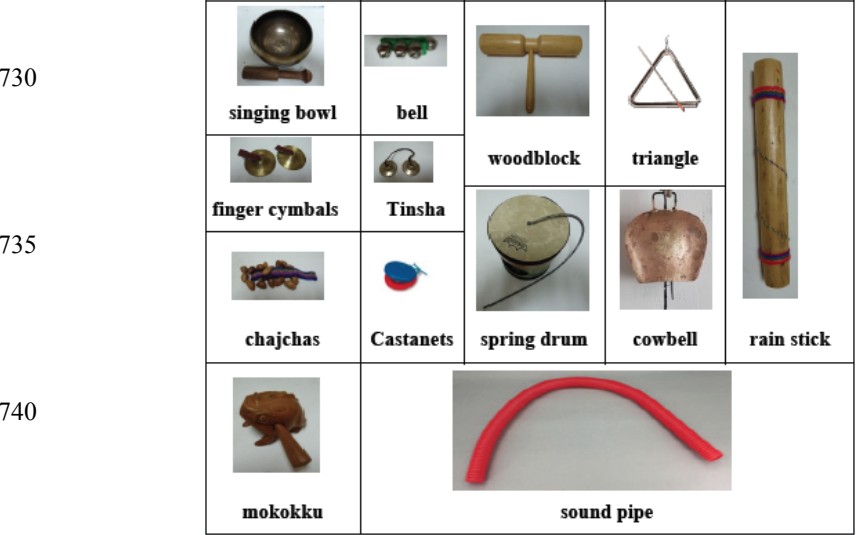




**Figure 10: Percussion instruments used in the activity for the university students.**





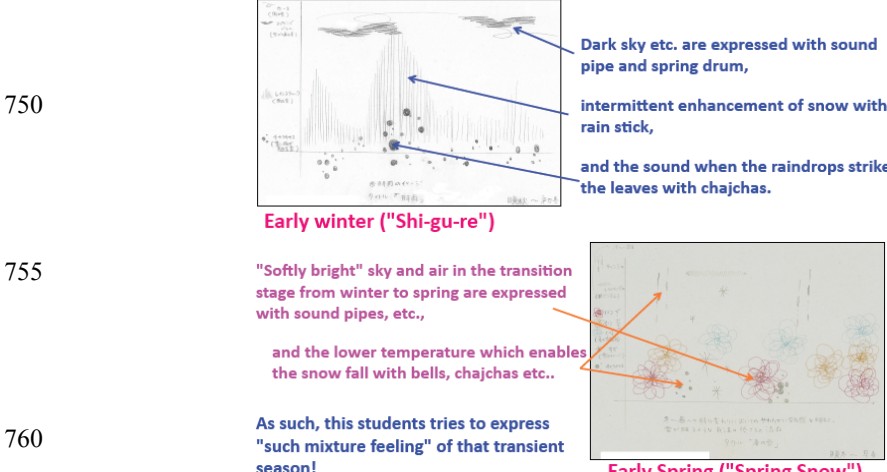

**Figure 11:** An example of the students' works on the music expression for early winter (upper panel) and early spring (lower panel) in the form of the graphic notation, after Kato et al. (2014). The titles by the student are "Shi-gu-re" and "Spring-Snow" for the works in the upper and the lower panels, respectively. Some explanation by the present authors was also added. The left and right edges of each figure indicate the beginning and end of the music piece, respectively.

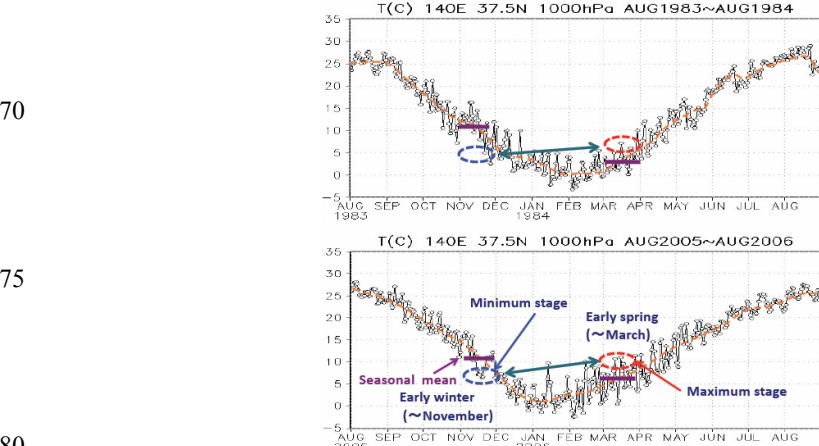

**Figure 12:** Examples of the day-to-day variation of daily mean surface air temperature (℃) around the eastern part of the Japan Islands (37.5°N/140°E) for the two years of 1983/1984 and 2005/2006 based on the NCEP/NCAR re-analysis data, modified from Kato et al. (2014). Names of the months are shown at the beginning of them.



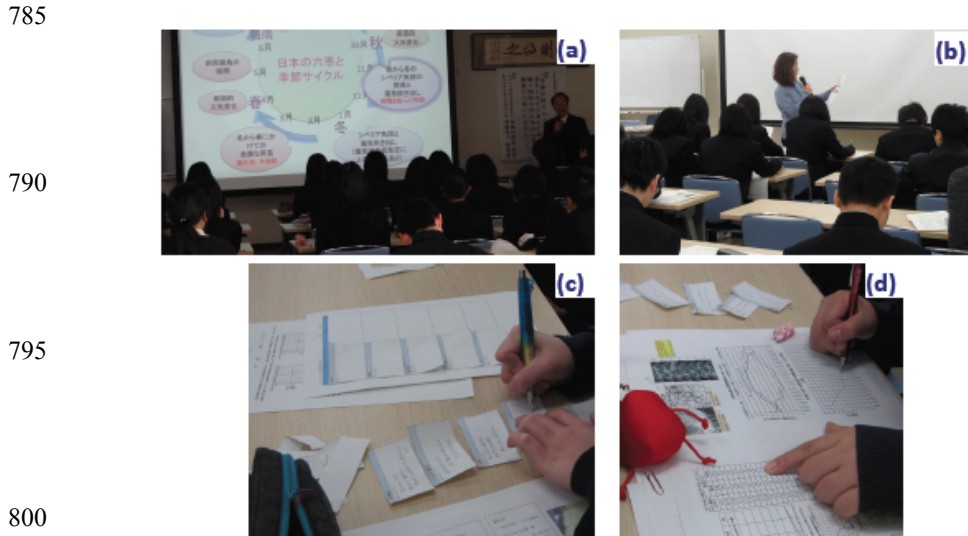

**Figure 13: Examples of the scenes of the lesson practice at a senior high school, after Kato et al. (2017a). The students are listening to the lecture on the outline of the climate in East Asia (Panel (a)) and the seasonal expression found in the Japanese school songs (Panel (b)). They are trying to arrange the several Japanese school songs which titles are written on the cards in the detailed seasonal order (Panel (c)). They are finding some features of the asymmetric seasonal progression around the Japan Islands by drawing the graph of the time sequence of 5-day mean air temperature there to compare with the other elements (Panel (d)).**








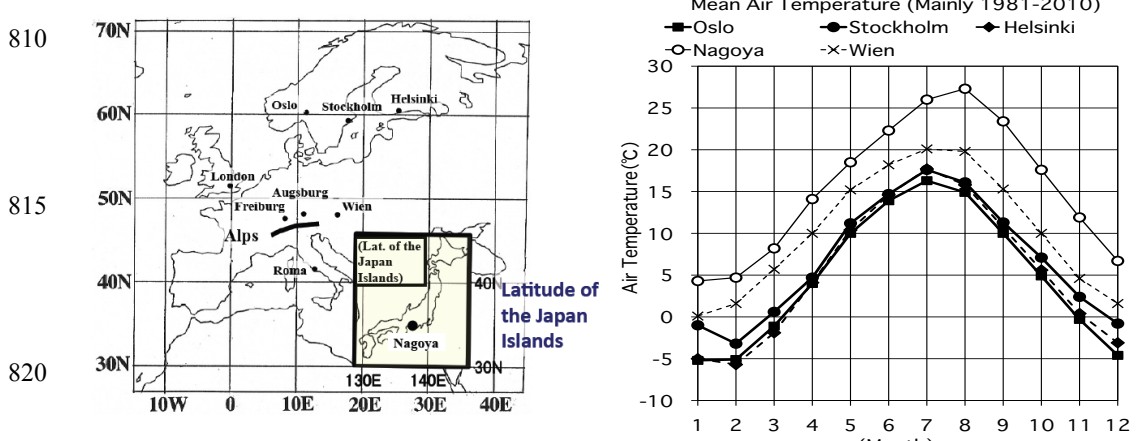

**Figure 14: Seasonal variations of monthly mean surface air temperature (℃) at Wien (Austria), Nagoya (Japan, 〜 300km westward from Tokyo in Fig. 4), Oslo (Norway), Stockholm (Sweden) and Helsinki (Finland) averaged for 1981 to 2010, except for**

**1982 to 1994 at Stockholm are shown in the right panel, after Kato et al. (2019). Locations of the cities are in the left one.**

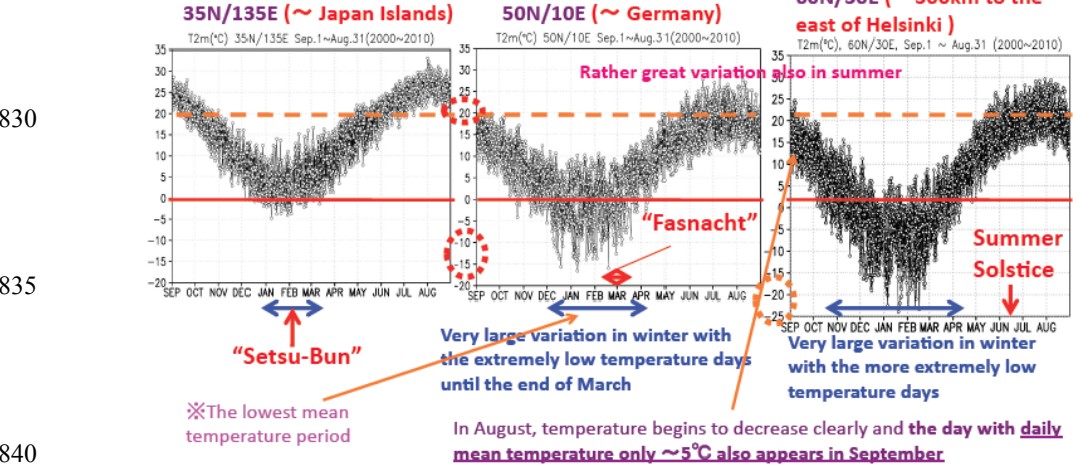




**Figure 15: Superpose of the sequences of daily mean surface air temperature (℃) for 11 years (Sep. 2000〜Aug. 2010) for around the central part of the Japan Islands (35°N/135°E) (left panel), central or southern part of Germany (50°N/10°E) (central one) and about 300km to the east of Helsinki (60°N/ 30°E) (right one), based on the NCEP/NCAR re-analyses data (modified after Kato et al. (2019)). Names of the months are shown at the beginning of them.**







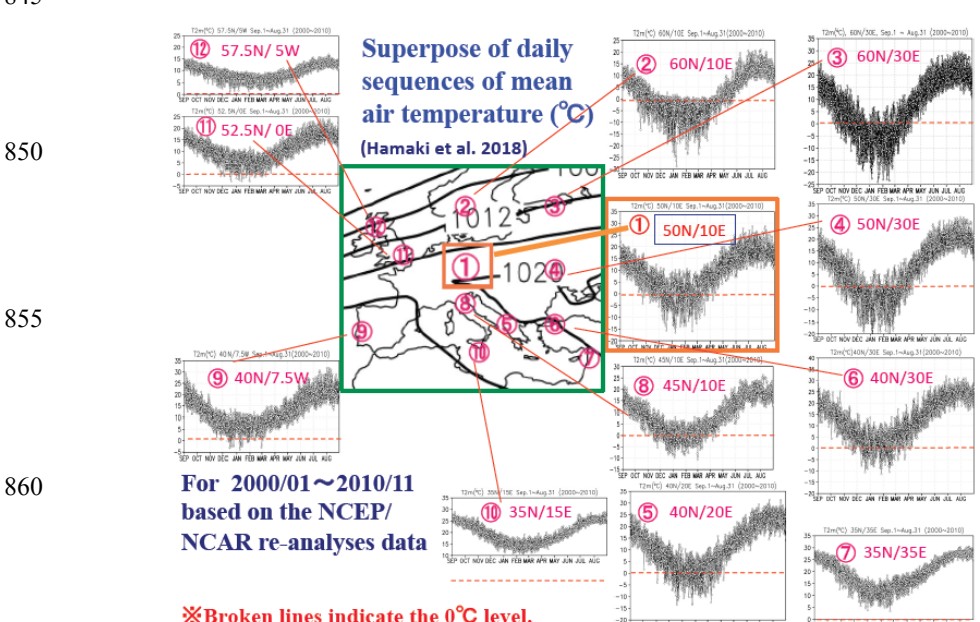




**Figure 16: Superpose of daily sequences of mean air temperature (℃) for 11 years (Sep. 2000～Aug. 2010) at various grid points, based on the NCEP/NCAR re-analyses data (after Hamaki et al. (2018)). Names of the months are shown at the beginning of them.**

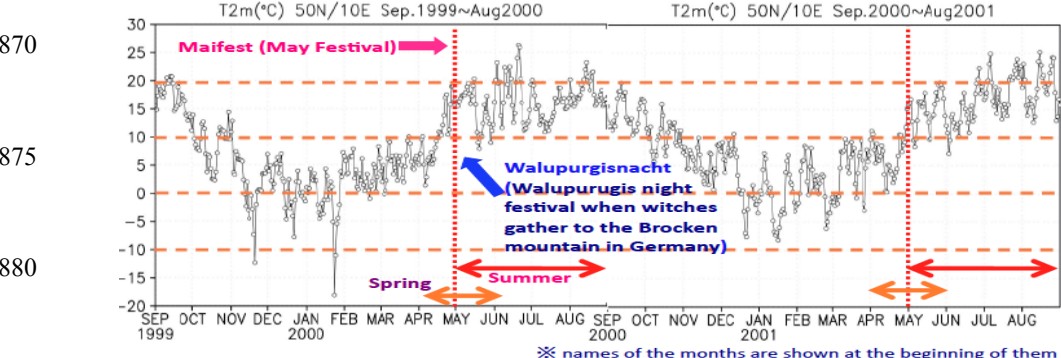




**Figure 17: Day-to-day variation of the daily mean surface air temperature (℃) around Germany (50°N/10°E) from Sep. 1999 to August 2001, based on the NCEP/NCAR re-analyses data (modified after Kato and Kato (2019)). Names of the months are shown at the beginning of them.**







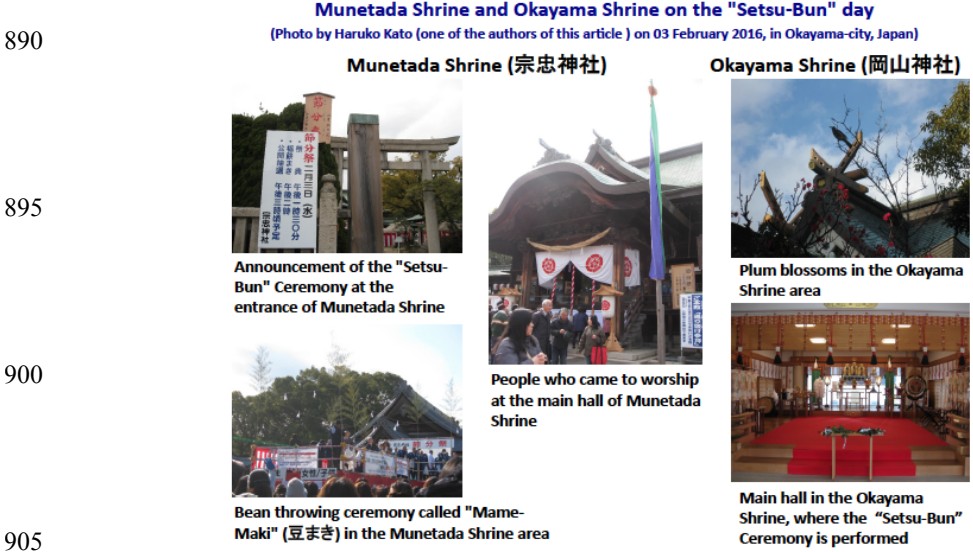

**Figure 18: Photographs of the "Setsu-Bun" scene at "Munetada Shrine" and "Okayama Shrine" in Okayama-city, Japan, by Haruko Kato, one of the present authors on 3 February 2016.**







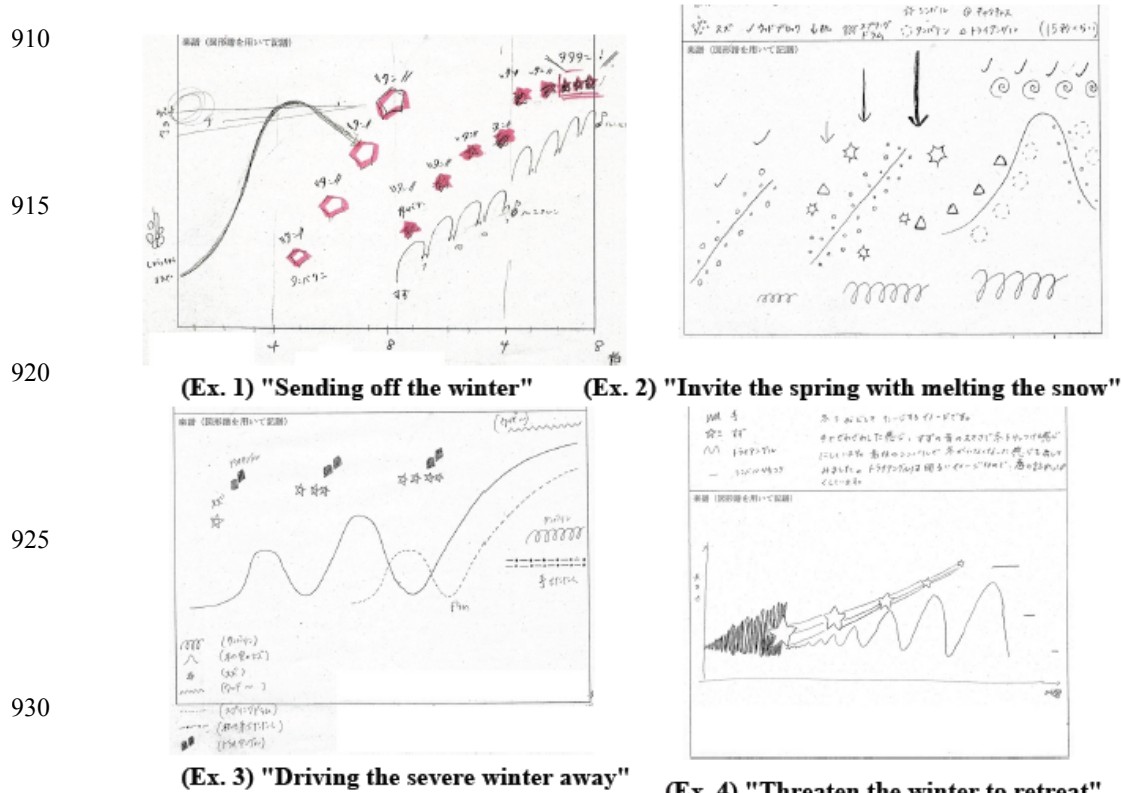

(Ex. 1) "Sending off the winter"    (Ex. 2) "Invite the spring with melting the snow"



(Ex. 3) "Driving the severe winter away"    (Ex. 4) "Threaten the winter to retreat"

**Figure 19: Examples of the students' composition works on their original "Fasnacht", described in the form of "graphic notation"**
**proposed by Morton Feldman (after Kato et al. (2017b)). The left and right edges of each figure indicate the beginning and end of the music piece, respectively.**

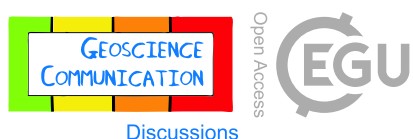

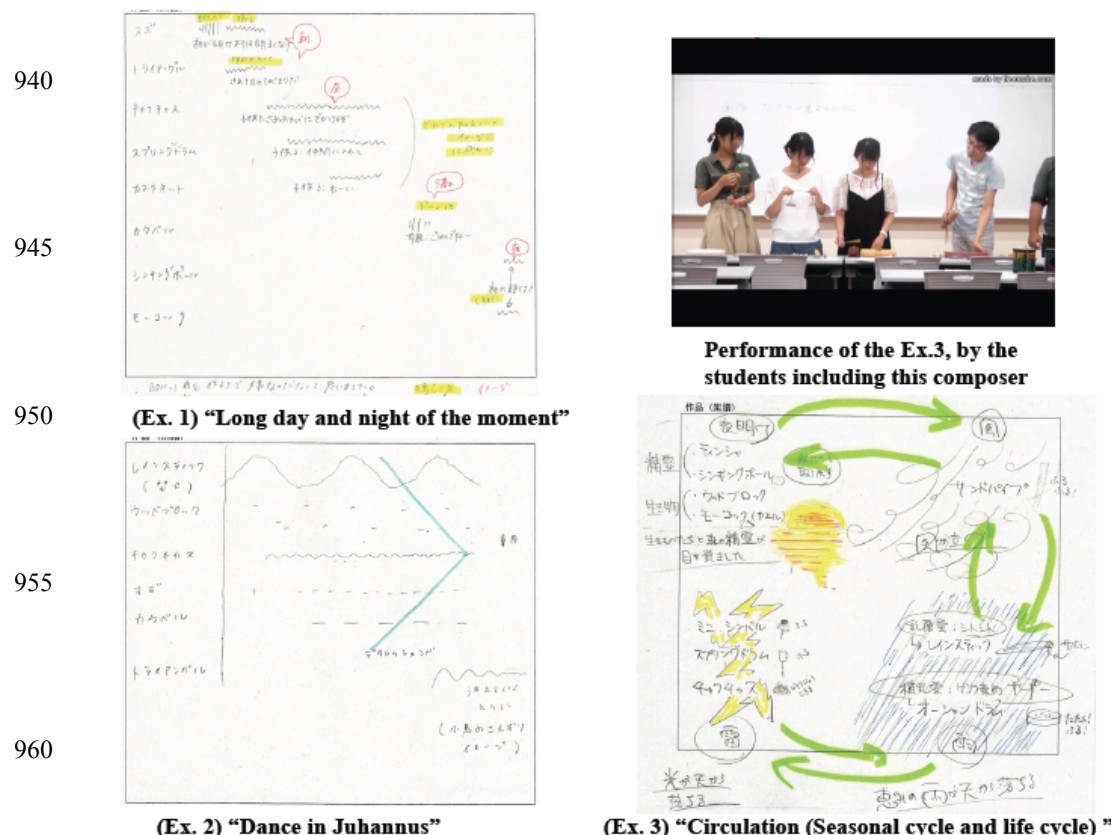

Figure 20: Examples of the students' composition works on Juhannus, described in the form of "graphic notation" proposed by Morton Feldman (after Kato et al. (2019)). The left and right edges of the works Ex. 1 and 2 indicate the beginning and end of the music piece, respectively. As for the Ex. 3, the progress of the piece is shown as the green arrows. Scene of the performance of the work Ex.3 is also shown in the upper right panel.