# Peer review of "Climate and music (Toward development of the interdisciplinary climate and cultural understanding education of ESD with special attention to the seasonal cycle and "seasonal feeling" around Japan and Europe)"

_Geoscience Communication, 2020_

## Referee Comment (RC1) · Anonymous Referee #1 · 21 May 2020

<Overall comments> This paper is a unique interdisciplinary attempt to integrate the knowledge on regional climate and music in order to educate some of the fundamental ESD literacies, such as thinking of various complex relations, diversity, understanding of heterogeneous others to university level students by the authors several educational activities conducted in Okayama prefecture, Japan. While climate, or more precisely in this article, climatic seasonal changes in different regions can be more easily treated as objective methodologies, how to treat its counterpart "music" or cultural matters in

objective scientific ways will be a hard task. The authors set three targeting issues in Japan and Europe. 1. Asymmetric seasonal feeling between early winter and early spring in Japan. 2. Seasonal feeling in Germany focusing on mid-winter "Fasnacht" for driving winter away. 3. Seasonal feeling in North Europe focusing on summer solstice festival. The selections of these specific seasonal events in each region are uniqueness of the authors perspective, and still there remain some ambiguity on how to express such feelings in scientific ways, their educational examples show at least partly the effectiveness of their approaches in university education. The reviewer is not familiar with the overseas situations, but this will be a quite unique educational attempt to understand the global diversity through climate and music. Therefore, it will be worth describing their educational results on this matter. There are, however, some of the issues from a view point of climatological seasonal cycles in this paper as pointed out below. Therefore, the paper needs minor/major revisions.

<Major comments> 1. Since this paper deals with regional climate, it will be better to add some references on the general regional climate features, for example, Arakawa and Toga (1969) for Japan, Schüepp and Schirmer (1977) for Germany, and Johannessen (1970) for Scandinavia. 2. Climatological six seasons in Japan, Maejima (1967) will be the first to identify these natural seasons, different from the traditional "four seasons" in mid-latitude regions. Arakawa and Tsuga (1969) also stated Japanese 6 seasons. Better to refer this paper on the Japanese climatological seasonal cycle. 3. The weather phenomena "Shi-gu-re" and the early spring snow "Awa-yuki" will be typically occurred in the semi-Japan Sea side climate situation (Suzuki, 1962), such as in Kyoto, but may rarely occur in the Pacific side climate such as in Okayama. This will be a big issue in regional climate recognition in Japan when educating this in Okayama. In addition, need to add the location of Okayama and Kyoto where most of "Waka" in Table 1 will be read in Fig. 5, and such information should also be elaborated on related with Table 1. 4. A number of regional different characteristics in seasonal cycles in Japan and Europe has been pointed out, but isn't it based on the different nature of the so-called "west coast" and "east coast" climate? Why winter

temperature variations are larger in Europe compared in Japan? 5. The authors seem to show the severity of winter in Germany or North Europe only by temperature. Is it enough to show the severe winter feeling in Europe? 6. Non-English words are better to be written in Italic.

References: Arakawa, H. and Taga, S. 1969. Climate of Japan. In Arakawa, H. ed. World Survey of Climatology, 8: 119-158, Amsterdam: Elsevier. Johannessen, T.W. 1970. The climate of Scandinavia. In Wallen, C. C. ed. World Survey of Climatology, 5, 23-79, Amsterdam: Elsevier. Maejima, I. 1967. Natural season and weather singularity in Japan. Geographical Reports Tokyo Metropolitan University 2: 77-103. Schüepp, M. and Schirmer, H. 1977. Climate of Central Europe. In Wallen, C. C. ed. World Survey of Climatology, 6: 3-73, Amsterdam: Elsevier. Suzuki H. 1962. Classification of Japanese climates. Geographical Review of Japan 35: 205–211 (in Japanese with English abstract).

---

## Referee Comment (RC2) · Anonymous Referee #1 · 21 May 2020

<Specific comments> 1. Figure 2: What are background colors indicated? If they have any meanings, they need to be added in the figure caption. 2. Figure 16: The contour lines seem to be sea level pressure, but are these contours needed? <Technical comments> 1. L51: Asian monsoon -> East Asian winter monsoon 2. L86: music -> Music (?) 3. L133: 35N -> 35°N 4. L164: In L164, it is 96. Which is correct? 5. L170: by Molton Feldman: Need reference, if available. 6. L498: Over -> over 7. L781: In figure title, it is shown as 1000 hPa temperature. Which is correct?

---

## Referee Comment (RC3) · Emilia Gómez (Referee) · 22 May 2020

General comments

The paper presents an interdisciplinary study relating climate, music and education, and contrasting these aspects in two different countries and cultures Japan vs Germany and Northern Europe. The paper is structured on a series of educational activities which include the explanation of climate concepts and cyclesand some exercises

where students relate climate to colors and generate music imitating certain climate-related processes, e.g. rain. The important climate concepts which are addressed are: (1) the asymmetric seasonal progression around Japan (autumn to midwinter and midwinter to spring); (2) the large day-to-day variation in mean surface temperature in winter in Germany, which is related to the Fasnacht; and (3) The traditional mid-upper event in Northern Europe, corresponding to the Summer Solstice, compared to summer climate in Japan. After including these climate concepts, the paper is structured around a set of educational activities carried out at the Faculty of Education where students were asked to composed a set of songs representative of these concepts.

The paper is very interesting and contributes to current state of the art, in particular given its interdisciplinarity, novelty and cross-cultural elements of comparison. However, I think the paper needs to be improved in terms of the description of the involved musical concepts, the methodology of the experiments and the presentation of the results in a rigorous way. I provide some comments for improvement from my expertise on music and computing.

Specific comments

I add here some comments for improving the paper:

- I think the structure of the paper could be better balanced. My advice would be to structure the paper in three main sections for each of the analyzed climate concepts: (1) description of the climate concept and characteristics; (2) description of associated music / colors with these events, including a list of musical pieces and art works; (3) presentation of the activity including the followed methodology; and (4) analysis of students results: study on the use of colors, instruments and temporal patterns, agreement between students, and conclusions on how climate concepts are related to musical ones.

- In general I think the paper focuses a lot on the climate concepts but it does not describes the relevant musical concepts, e.g. which songs are linked to the climate

events in each culture, which are the properties of these songs (emotion, instruments, tempo). We miss also a description of the musical instruments used in the study, and a description of other musical concepts considered (e.g. tempo and timbre). It is very important to describe this in order to fully understand the links and results of the experiments.

- 2.2.1: please describe better the expression of the four seasons (Itten 1961) and relate it to the results in Figure 9 commented in 2.2.2.

- I miss a more rigorous analysis of the results of the three experiments, as Figures 11-19 and 20 only present and comments few examples. We would need to understand the results for all students following the same methodology, and see if we can get some conclusions on the general findings and individual differences in the way they related music, climate and emotions. It would then be interesting to analyze the agreement between the students, and if they select the same instruments and temporal patterns in the same way. I advise the authors to consider and relate to the large corpus of literature relating music to emotions that one would need to cite here, such as: [1] Juslin, P. (2019). Musical emotions explained. Oxford University Press. [2] Meyer, L. (1956). Emotion and the Meaning of Music. Oxford University Press. [3] Eerola, T., and Vuoskoski, J.K. (2013). A review of Music and Emotion studies: approaches, emotion models, and stimuli. Music Perception: An Interdisciplinary Journal, 30, 307-340. [4] Balkwill, L.L. and Thompson, W.F. (1999). A cross-cultural investigation of the perception of emotion in music: psychophysical and cultural cues. Music Perception, 17, 43-64.

Technical corrections

- I suggest the authors to add some complementary material to the paper which that can be useful for future educational and research activities on the area. I suggest to include a list of representative songs from Japan and Germany representative of each climate season

- Figures 19 and 20 are difficult to ready. I would suggest to add a description in the text.

As a conclusion, I think the paper topic is very interesting and there is a lot of work behind it, but I think it would benefit a lot from a change in the structure and a more detailed description of the results, so that the paper is self-contained for publication.

---

## Referee Comment (RC4) · Louise Arnal (Referee) · 26 May 2020

General comments.

This paper explores the fusion of climate science and art (music and visual arts) as means to promote ESD (education of sustainable development) literacy, as well as to further the understanding of different regional climates and cultures. They present an overview of educational activities (previously reported in several Japanese books and

papers) in which students are presented with seasonal climate features of Japan, N. Europe and Germany, and have to reflect on these using visual arts and music. The authors reflect on and highlight differences between the three region's climate features. The paper tackles the very essential topic of climate education, especially in these times of changing climate. I particularly enjoyed reading about the educational activities carried out by the authors and learning about some key climate features in Japan and Europe. In my opinion this paper is more than suitable for publication in Geoscience Communication, for the special issue's theme of 'Five years of Earth sciences and art at the EGU (2015–2019)' after revision. Please find below a collection of comments that I would like to raise, and which I would encourage the editor and the author to consider prior to publication:

- I think that the main point of this paper is watered down and the paper would benefit from clarifying this: climate features are difficult to communicate to promote ESD literacy without the use of art, narratives and references to individual and cultural feelings. Additionally, the paper is centered around two main aims: 1) to promote ESD (education of sustainable development) literacy, and 2) to further the "Understanding of Heterogeneous others". This needs to be clarified throughout the paper and should be used as a basis to discuss the results in the Discussion section.

- While the paper mentions music very clearly, the use of visual arts in the activities is not acknowledged. Please consider adding some literature on visual arts and science in the introduction and mentioning the use of visual arts in the abstract and Discussion.

- Title. In my opinion the title is too long and therefore looses in catchiness. I didn't know what ESD meant before reading this paper, and this might need to be spelled out in the title. Education of ESD seems to be repetitive. Consider using a colon instead of the parentheses, as this is rather unusual for titles.

- Abstract. The abstract could benefit from a few clarifications: Why is this work important? I understand why it is but think you need to clarify the aims/importance of your

work with regards to these points/the wider context. What does this paper investigate and show? Please also highlight some of the results of this paper here.

- Introduction. 1) The introduction needs to give a wider overview of the context in which your work situates itself and acknowledge other works on similar topics (this will be particularly helpful for readers like me who are not very familiar with the literature in this field). While some parts are in my opinion too detailed and distract from the main message (e.g. P2 L54-73, P3 L74-82). Here are some guiding questions which will hopefully be helpful to the authors to clarify the framing of the introduction: What is ESD and why is it important? What are the benefits of linking music and climate science? (In other words, what are some challenges that the fusion of music and climate science can help tackle?) How can their fusion help improve ESD literacy in different regions and for different seasons? What are you proposing to do in this paper to tackle these challenges? 2) Please also introduce some of the keywords used throughout your paper in the introduction, such as "seasonal feeling". 3) Please explain briefly here why you have chosen to focus on Japan, N. Europe and Germany.

- Results. 1) Your paper could gain in clarity by following a more rigorous structure, as also mentioned by reviewer Emilia Gomez. Please consider moving these sections to an overarching section 2, as well as splitting results for N. Europe and Germany. 2) Your results would benefit from additional explanations of some of the music concepts you for example mention briefly on P10 L311-312 ("changing loudness", "tone of the sounds") and a description of the instruments used (which I am for the most part not familiar with), as well as some of the climate keywords (e.g. sensible vs latent heat). Additionally, your results need further explanations and analyses of the links made by students between colours - music instruments and concepts (e.g. tone) - climate features. For example, I found your analysis of the results of activity 2 in section 2.2.2 great, but the analyses of results for Germany and N. Europe are rather light. Did most students use the same instruments/colours/... to convey the same feature? What does this choice translate in their understanding? If they used different

instruments/colours/. . ., why could that be? For example, could you please illustrate the point you make on P7 L194-198 with activity outputs? 3) Please also clarify the methods used (Johannes Itten's exercise and the "graphic notation" from Morton Feldman). 4) Was there any time to discuss the activity results with the students, or is your analysis of the results only based on your personal observations? Please clarify this in the paper. 5) I found it difficult to jump between the different figures you mention in section 3.1 and 3.2. 6) While reading through more and more of the results, I found it hard to remember which climate features you had used for each region. It might be helpful to make a table of key climate features per region (Japan, Germany and N. Europe) that the readers can refer to throughout the paper. 7) Why are the results of activities 3 not presented for Germany and N. Europe? I found these very interesting for Japan. Mention that you are focusing on the music activities if that is indeed the case.

- Discussion. The discussion could benefit from: tying your results back with some of the wider context literature you (will) mention in the Introduction, and further reflections on the two main points of your work: how have these activities helped with ESD literacy and with "Understanding of Heterogeneous others"? Here are a few more specific questions (as well as a couple more in the specific comments) which it would be great to see discussed here: how transparent is art to the changing climate we are observing/feeling? (e.g. Have you seen a change in music about climate through time which could at least partly reflect climate change? How could art help in these challenging changing times?) Can the activities outputs indeed communicate seasonal feelings to people who have no prior knowledge of the regional climate features at stake? How successful were these activities according to you? (e.g. What were some of the challenges? What would you improve on/do differently for future implementations?)

Specific comments.

- P1 L9-12: This part of the abstract is a repetition word for word of the introduction P1 L25-28. Please rephrase.

- P1 L9-11 and L25-27: Are these results based on an analysis you carried out in the paper/found in other published material? You need some material to back this up I think. Please consider rephrasing and adding references.

- P1 L12 and L28: Please clarify what you mean by "the other background".

- P1-2 L28-30: I understand what you mean but it is not very clearly worded. Please consider clarifying.

- Fig. 1: The collaboration is what you focus on? This needs to be said explicitly in the introduction.

- P2 L32-33: It is not clear to me why cultural understanding is a part of climate education. Is that a point you are trying to make in this paper, or that you have made in other published materials? Please clarify.

- P2 L35: Where is the term "Fundamental ESD Literacy" from (add reference) and what does it mean exactly? Same with "Understanding of Heterogeneous others" on P2 L37.

- P2 L43-44: Could you please add at least one reference for this statement.

- P2 L54-55: Please rephrase, I am not sure to understand the meaning of this sentence.

- P3 L69-73: I would move this to the Discussion section and explore it further there. Has it indeed helped with ESD literacy with regards to these climate features? In what ways?

- P3 L92-96: This distracts from the mains points of the Introduction. I would remove these and instead say that your paper's structure follows that of Kato & Kato (2019).

- P4 L99-101: Before introducing the structure of your paper, please say that you focus on a selection of climate features in these activities. Please clarify in which paper sections these 3 parts are discussed.
- P4 L103-107: I suggest to move this paragraph to the relevant section 2.

- P4 L112-113: This is however not observed in the left panel, which is a bit confusing. Please clarify that they show the temperature observed at different altitudes.

- P4: I found Fig. 5 and 6 complicated to read. It would be great if you could guide the readers through these in the text.

- P5 L147: Did you organize this activity in a different season than the one explored during the activity on purpose? It would be interesting to read your reflections on the choice of the activity date in the Discussion. In your opinion, would it have impacted the results/success of the activity if run during another season?

- P5 L155-156: Please clarify what ages these grades correspond to (also for the other sections).

- P5 L160: Please clarify what "96 coloured papers" refers to. Did you offer the students an array of 96 different colours to choose from? Why is it 93 on P6 L164?

- P6 L164: What do you mean by "which color combination is best"? Do you mean the combination which most accurately represents this climate feature for the student? I am wary that it may come across as: which colour combination is the most aesthetically pleasing, which isn't the result you are looking for here.

- P6 L179: Please give examples of "cool colors" for readers less familiar with this concept.

- P7 L199-201: Please clarify.

- P7 L202-204: Link this observation with the outputs of Fig. 9, where there are clearly different personal emotions at play as not all outputs are the same.

- P7 L208-210: That would be an interesting follow up of this work and might be good to add and discuss in the Discussion section.

- P7 L211-215: I would shorten this paragraph and remove Fig. 13 as you already have a lot of figures.

- P8 L232: Is "Wien" used as a proxy for Germany in Fig. 14 (and others)? Please clarify in the text and figures captions if so.

- P8 L237: Which panel number are you referring to on Fig. 16. Please clarify in the text and provide labels for panel numbers in the caption.

- P8 L237 and L247: Please clarify which location "there" refers to here.

- P8 L238-239: I find this hard to see in Fig. 15. It might help if you could quantify the maximum summer day-to-day variations.

- P8 L243-247: This is a very interesting contrast. Did you explore this with the music activity?

- P8 L247-248: Please clarify.

- P9 L259-261: I find this hard to see in Fig. 14.

- P9 L263-264: I find this hard to see in Fig. 15.

- P9 L266-267: Do you mean by this the day to day variations?

- P11 L343-347: Please add the figure numbers relating to the titles for which they are available. Where is ex1 situated amongst these two categories?

- P12 L368-370: Could you please illustrate this statement with an output of one of the activities?

- P13 L401-410: Please clarify this paragraph and illustrate with specific examples from your results.

Technical corrections.

- P1 L12: "inversely" doesn't fit here.

- P1 L14 and other instances: I would suggest changing "By the way" to another more formal term.

- P2 L29 and multiple other instances: change "and so an" to "etc." or "among others".

- P2 L41-42: please clarify what you mean by "as if they have been there since before".

- P2 L46: "called".

- P3 L89: "were obtained" instead of "were got".

- P4 L99 and other instances: What do you mean by "The asymmetric seasonal march"? The transition?

- P5 L139 and multiple other instances: I am not familiar with the term "lesson practice" and would suggest changing this to, for example, "activity".

- P6 L240: "the month of May".

- P6 L256: "have done many things".

- P9 L258: "dance together throughout the night".

- P10 L289: "explanations of the works given by the student".

- P10 L297: "the sky has completely fallen down".

- P10 L298: "the lively coming spring".

- P11 L346: "this category are".

- P12 L355: What do you mean by "Mokokku", "flogs" and "mysterious singing ball"?
* * *

---

## Referee Comment (RC5) · Sun-Kyung Lee (Referee) · 30 May 2020

General comments : This study intends to link climate education with music education through very interesting pedagogical approaches regarding climate. Therefore, it is suitable for the special issue of this journal, Geoscience Communication. However, it also has a couple of issues which need to be discussed and revised to be more meaningful. These are some comments to consider when this article is revised.

[Figure]

Specific comments : First of all, climate or seasonal changes are usually very closely related to an individual's emotions, life or culture, but many people do not feel it well or notice the connection of them in everyday life. In this regard, this paper attempted to study various scientific geographic characteristics related to climate in Japan and Europe including Germany, to think about the feelings of climate at specific seasons, and to express them with color or music, to appreciate and discuss them. Through this process, it is thought that the influence of climate on human emotions and life can be well understood. However, it may be necessary to clarify and distinguish whether changes in a specific period noted in this paper are climate phenomena that can be explained as characteristics of a longer period or meteorological phenomena caused by seasonal variations.

Second, meteorological variations and climate change have a major impact on people's lives. They can make psychological changes of people and it can change the way people act. Therefore, the ability to be sensitive on climate change to detect or link the change in life or society, is essential for a sustainable future. In terms of ESD (education for sustainable development), it is important to understand the complexity of climate itself. However, it is more important to address the complexity of climate change issue, which is not clearly visible in this article.

The third is the pedagogical aspect for climate learning. Authors use various teaching and learning methods including traditional lectures with explanations, graphs, tables, etc., and more innovative methods with project-based learning including composition, playing with instruments, expressions of feelings, explaining etc. The use of these various pedagogical approaches gives students a variety of approaches to climate topics. Therefore, they can experience various aspects of this topic. Also in this process, not only science and music, but also art, Japanese language, social studies, geography learning are involved. Therefore, Fig. 1 of p 19 will need to be revised.

The fourth aspect is related to the lesson study mentioned in the third part. Although this covers lessons, it is difficult to say that it is a lesson study that is generally practiced

in Japan or other countries. Authors might want to reconsider the use of the term.

---

## Author Comment (AC1) · 15 Oct 2020

Thank you very much for your careful reading of our manuscript and very helpful comments and suggestions. We apologize for our not so quick response due to the unusual situation in our university affected by the Coronavirus epidemic and its prevention.

In the following authors' response to the reviewers' comments, we will italicize the comments, and add our answers below. In the answers, page and line numbers and figure

numbers are based on the previous manuscript uploaded as the discussion paper.

<Overall comments> This paper is a unique interdisciplinary attempt to integrate the knowledge on regional climate and music in order to educate some of the fundamental ESD literacies, such as thinking of various complex relations, diversity, understanding of heterogeneous others to university level students by the authors several educational activities conducted in Okayama prefecture, Japan. While climate, or more precisely in this article, climatic seasonal changes in different regions can be more easily treated as objective methodologies, how to treat its counterpart "music" or cultural matters in objective scientific ways will be a hard task. The authors set three targeting issues in Japan and Europe. 1. Asymmetric seasonal feeling between early winter and early spring in Japan. 2. Seasonal feeling in Germany focusing on mid-winter "Fasnacht" for driving winter away. 3. Seasonal feeling in North Europe focusing on summer solstice festival. The selections of these specific seasonal events in each region are uniqueness of the authors' perspective, and still there remain some ambiguity on how to express such feelings in scientific ways, their educational examples show at least partly the effectiveness of their approaches in university education. The reviewer is not familiar with the overseas situations, but this will be a quite unique educational attempt to understand the global diversity through climate and music. Therefore, it will be worth describing their educational results on this matter. There are, however, some of the issues from a view point of climatological seasonal cycles in this paper as pointed out below. Therefore, the paper needs minor/major revisions.

Thank you very much for your showing an interest in our manuscript. We will revise our manuscript after the following manner by the consideration your comments and suggestions as below.

<Major comments> 1. Since this paper deals with regional climate, it will be better to add some references on the general regional climate features, for example, Arakawa and Taga (1969) for Japan, SchuÌ€Lepp and Schirmer (1977) for Germany, and Johannessen (1970) for Scandinavia.

Thank you very much for your suggestions. We will add these references on the general regional climate for three regions.

2. Climatological six seasons in Japan, Maejima (1967) will be the first to identify these natural seasons, different from the traditional"four seasons" in mid-latitude regions. Arakawa and Taga (1969) also stated Japanese 6 seasons. Better to refer this paper on the Japanese climatological seasonal cycle.

Thank you very much for your information. We will refer to Maejima (1967), Arakawa and Yaga (1969), on the Japanese climatological seasonal cycle with the "6 seasons".

3. The weather phenomena "Shi-gu-re" and the early spring snow "Awayuki" will be typically occurred in the semi-Japan Sea side climate situation (Suzuki, 1962), such as in Kyoto, but may rarely occur in the Pacific side climate such as in Okayama. This will be a big issue in regional climate recognition in Japan when educating this in Okayama. In addition, need to add the location of Okayama and Kyoto where most of "Waka" in Table 1 will be read in Fig. 5, and such information should also be elaborated on related with Table 1.

After your suggestion, we will add the locations of Kyoto and Okayama with the light modification in the right panel of Fig. 5, together with adding the figure's caption as below. And in the text, it will be also mentioned that Kyoto is identified as the semi-Japan Sea side climate situation (Suzuki, 1962) and the "Shi-gure" may rarely occur in the Pacific side climate such as in Okayama where our lesson practices were performed. We will also add the statement in the caption of Table 1 to refer to the location of Kyoto in Fig. 5.

In addition, we would also like to note that, although there may be many students in Okayama University who have not experience either the Japan Sea side climate or the semi-Japan Sea side climate, they have learned them at least in junior high school. Thus we think that such phenomena would be not so inadequate for our lesson practice.
4. A number of regional different characteristics in seasonal cycles in Japan and Europe have been pointed out, but isn't it based on the different nature of the so-called "west coast" and "east coast" climate? Why winter temperature variations are larger in Europe compared in Japan?

Thank you very much for your comments. As for the mean values of the climatological variables, the different nature of the so-called "west coast" and "east coast" climate would be the important basis, including the role of the winter mean Icelandic low in the "west coast" climatic situation. It should be also noted that intraseasonal variation of the Icelandic low is very large. While the rather warmer days persists as the intraseasonal variation when the Icelandic low shifts near the European side, due to the lager warm air advection, the temperature sometimes shows extremely low when it is located rather westward from the Europe, according to the case study of our group as briefly introduced in 3.3 (L232-235 in P8) (Hamaki et al. 2018, although written in Japanese). However, their analysis is limited to the case for 2000/2001 winter and the mechanism for the appearance of the extremely low temperature periods other than the absence of the warming effect of the eastward shift of the Icelandic low. Thus we only stated as the original manucsript. However, the larger day-to-day variation of the air temperature around Germany and Northern Europe than in Japan seems to be very interesting also at the viewpoint of the relation to the "seasonal feeling". As for the emotional sense, various subjective acceptance of the natural environment would be possible, depending on the inner situation of a person. For example, they might feel the seasonal coldness not only by the mean temperature but also by the intermittent appearing the rather low temperature situations in some regions. Although the objective relationship between such day-to-day variation and the "seasonal feeling" seems to be hard to be clarified by the authors' special fields, it would be very helpful to understand/imagine the natural environmental around Germany and Northern Europe as an important factors relating to the "seasonal feeling" of the "severe winter" there, at the first step for the lesson practice on the interdisciplinary cultural understanding education.

5. The authors seem to show the severity of winter in Germany or North Europe only by temperature. Is it enough to show the severe winter feeling in Europe?

The authors also do not think it enough to show that and it is needed to examine the effects of the other factors. However, we thought that only paying attention to the day-to-day variability of the temperature as well as its climatological mean would give an unexpected fresh finding to the students and we showed only the temperature for understanding the severe winter situation in our lesson practice.

6. Non-English words are better to be written in Italic.

Thank you very much. We will do so.

ïijŁPlease also refer to the supplement pdf file including some figures together with the above text.

Please also note the supplement to this comment:
https://gc.copernicus.org/preprints/gc-2020-18/gc-2020-18-AC1-supplement.pdf

---

## Author Comment (AC2) · 15 Oct 2020

Thank you very much for your careful reading of our manuscript and very helpful comments and suggestions. We apologize for our not so quick response due to the unusual situation in our university affected by the Coronavirus epidemic and its prevention.

In the following authors' response to the reviewers' comments, we will italicize the comments, and add our answers below. In the answers, page and line numbers and figure

numbers are based on the previous manuscript uploaded as the discussion paper.

<Specific comments> 1. Figure 2: What are background colors indicated? If they have any meanings, they need to be added in the figure caption.

The background colors do not have any climatological meanings. However, for visual readability, some background colors are indicated. We will add such statement in the figure's caption.

2. Figure 16: The contour lines seem to be sea level pressure, but are these contours needed?

The contour lines were for the sea level pressure in climatological January mean. However, as the reviewer points out, the contours were deleted and the figure will be replaced as follows.

<Technical comments> 1. L51: Asian monsoon -> East Asian winter monsoon 2. L86: music -> Music(?) 3. L133: 35N -> 35°N 6. L498: Over -> over

Thank you very mich. We will do so on 1. to 3. and 6.

4. L164: In L164, it is 96. Which is correct?

"93 colors" is correct. We will change into the correct one.

5. L170: by Molton Feldman: Need reference, if available.

"Graphic notation" is also sometimes used in the modern music as well as the conventional notation called "mensural notation" or "staff notation". Thus presenting the references on the general explanation of the "graphic notation", in the research sense, seems to be not so easy for us. Instead, we are thinking to present a dictionary of music, e.g., "Das Grosse Metzler Musiklexikon", as a reference.

7. L781: In figure title, it is shown as 1000 hPa temperature. Which is correct?

Thank you very much. 1000hPa is correct. For convenience of the preparation of
the figures at the lesson practice about 8 years before, we used the temperature at 1000hPa as the substitute for those at the surface level in Kato et al. (2014). Thus such statement will be added to the caption of Fig. 12 as follows, together with the other necessary parts.

Figure 12: Examples of the day-to-day variation of daily mean air temperature at 1000hPa level (°C) around the eastern part of the Japan Islands (37.5°N/140°E) for the two years of 1983/1984 and 2005/2006 based on the NCEP/NCAR re-analysis data, modified from Kato et al. (2014). Names of the months are shown at the beginning of them. The data at this level were used as the substitute for those at the surface level in Kato et al. (2014).

Please also note the supplement to this comment: https://gc.copernicus.org/preprints/gc-2020-18/gc-2020-18-AC2-supplement.pdf

---

## Author Comment (AC3) · 15 Oct 2020

Thank you very much for your careful reading of our manuscript and very helpful comments and suggestions. We apologize for our not so quick response due to the unusual situation in our university affected by the Coronavirus epidemic and its prevention. In the following authors' response to the reviewers' comments, we will italicize the comments, and add our answers below. In the answers, page and line numbers and figure

numbers are based on the previous manuscript uploaded as the discussion paper.

General comments: This study intends to link climate education with music education through very interesting pedagogical approaches regarding climate. Therefore, it is suitable for the special issue of this journal, Geoscience Communication. However, it also has a couple of issues which need to be discussed and revised to be more meaningful. These are some comments to consider when this article is revised.

Thank you very much for your interest in our study and your valuable comments. Our response to your specific comments is as follows.

First of all, climate or seasonal changes are usually very closely related to an individual's emotions, life or culture, but many people do not feel it well or notice the connection of them in everyday life. In this regard, this paper attempted to study various scientific geographic characteristics related to climate in Japan and Europe including Germany, to think about the feelings of climate at specific seasons, and to express them with color or music, to appreciate and discuss them. Through this process, it is thought that the influence of climate on human emotions and life can be well understood. However, it may be necessary to clarify and distinguish whether changes in a specific period noted in this paper are climate phenomena that can be explained as characteristics of a longer period or meteorological phenomena caused by seasonal variations.

In the activities referred to by this review article, we paid attention to some interesting characteristics of the climatological mean seasonal cycles presented with the 30-year mean values and their "standard variability"for that period, although some figures of mean features including the variability for about 10 years are shown as for Germany and Northern Europe (the climate normals of various meteorological elements provided by the Japan Meteorological Agency are the 30-year means). In addition, in order to illustrate the features of the day-to-day variation in the seasonal cycle, the time series for some specific years were also shown (e.g., Figs. 12 and 17). Thus we will try to

add a very short explanation of that in the text when Figs. 12 and 17 appear.

We would like to mention that magnitude of day-to-day variation of a meteorological element is also an important indicator of the "mean seasonal state", although its extreme values are sometimes associated with the climate change. Thus we also used the day-to-day variations as a study material for understanding the detailed but important "standard seasonal features" in various regions and seasons.

Second, meteorological variations and climate change have a major impact on people's lives. They can make psychological changes of people and it can change the way people act. Therefore, the ability to be sensitive on climate change to detect or link the change in life or society, is essential for a sustainable future. In terms of ESD (education for sustainable development), it is important to understand the complexity of climate itself. However, it is more important to address the complexity of climate change issue, which is not clearly visible in this article.

I really agree your opinion. In the education on Climate Change including the Climate Action in ESD, it is very important to understand how the climate has been changing, how kind of regional climate response is likely to occur in the future, and how complex the climate change issue is, and so on. However, the change in the future climate can be detected not only by the deviation from the present one but also by the change in the seasonal cycle pattern, i.e., changes in the seasonal maximum and minimum of the "mean" meteorological elements, their maximum and minimum phases, duration of their maximum and minimum phases, their "mean" day-to-day or year-to-year "variability" in each season, and their combination of these elements. We could tentatively call such changes the "distortion of the seasonal progression". Especially in middle and higher latitudes, the seasonal cycle of the climate system is clearly seen but shows rather different features from region to region. Thus, in detecting how the climate has been changing or understanding how it is likely to change in the future, it seems to be an effective way to pay attention to the "distortion of the seasonal progression". In order to do so, understanding of the complexity of the climate itself including the detailed

seasonal cycles differently from region to region would be necessary for that basis. If the students understand such detailed seasonal cycles at least for a few regions more deeply, they would be also able to feel more sensitively that the climate is somewhat different from before, which would help for detecting the climate change also scientifically. This is another reason why we focus mainly on the detailed seasonal cycle itself in our activity, as well as for providing the students' opportunity for promoting the ESD literacy.

The third is the pedagogical aspect for climate learning. Authors use various teaching and learning methods including traditional lectures with explanations, graphs, tables, etc., and more innovative methods with project-based learning including composition, playing with instruments, expressions of feelings, explaining etc. The use of these various pedagogical approaches gives students a variety of approaches to climate topics. Therefore, they can experience various aspects of this topic. Also in this process, not only science and music, but also art, Japanese language, social studies, geography learning are involved. Therefore, Fig. 1 of p 19 will need to be revised.

Thank you very much for your suggestions. We will revise Fig. 1 as follows.

Please also note the supplement to this comment:
https://gc.copernicus.org/preprints/gc-2020-18/gc-2020-18-AC3-supplement.pdf

---

## Author Comment (AC4) · 8 May 2021

Authors' Response to RC3 by Emilia Gómez (Referee)

Thank you very much for your careful reading of our manuscript and very helpful comments and suggestions. We apologize for our not so quick response due to the unusual situation in our university affected by the Coronavirus epidemic and its prevention. In

the following authors' response to the reviewers' comments, we will italicize the comments, and add our answers below. In the answers, page and line numbers and figure numbers are based on the previous manuscript uploaded as the discussion paper.

General comments The paper presents an interdisciplinary study relating climate, music and education, and contrasting these aspects in two different countries and cultures Japan vs Germany and Northern Europe. The paper is structured on a series of educational activities which include the explanation of climate concepts and cycles and some exercises where students relate climate to colors and generate music imitating certain climate related processes, e.g. rain. The important climate concepts which are addressed are: (1) the asymmetric seasonal progression around Japan (autumn to midwinter and midwinter to spring); (2) the large day-to-day variation in mean surface temperature in winter in Germany, which is related to the Fasnacht; and (3) The traditional mid-upper event in Northern Europe, corresponding to the Summer Solstice, compared to summer climate in Japan. After including these climate concepts, the paper is structured around a set of educational activities carried out at the Faculty of Education where students were asked to compose a set of songs representative of these concepts. The paper is very interesting and contributes to current state of the art, in particular given its interdisciplinarity, novelty and cross-cultural elements of comparison. However, I think the paper needs to be improved in terms of the description of the involved musical concepts, the methodology of the experiments and the presentation of the results in a rigorous way. I provide some comments for improvement from my expertise on music and computing.

Thank you very much for your showing an interest in our manuscript and for your giving us many valuable comments. We will revise our manuscript after the following manner by the consideration your general and specific comments and suggestions as below.

We think that your suggestion on the description of the involved musical concepts, the methodology of the experiments and the presentation of the results would be very important, especially in evaluating the effects of the activities in somewhat statistical and

objective way. However, construction of a kind of method for achieving a certain goal such as acquiring ability on musical expressions is somewhat different from the purpose of this research. In the ESD (Education for Sustainable Development) Teacher Education, it would be also very helpful for promoting the university students' "Fundamental ESD literacy" as stated in the text (L35-36 in P2) through their accumulation of the experiences of interactive consideration between "careful examination of the climate data" and "deeper appreciation of the seasonal feeling" expressed in the music works or traditional events (L40-42 in P2). Such activity would contribute not only to promoting the students' "Fundamental ESD literacy" as the pre-service teacher training but also to their ability to discover the various possibility in their development of such kind of interdisciplinary study materials as those relating to "climate and music" when they become teachers. Thus our studies have firstly focused on the various aspects of the climate and seasonal cycle including their variability as an important background of the cultural generation such as music and then we have examined the music works etc. and climate and seasonal features, in order to propose the various possibility for the study materials (not directing to choosing only one way of them). In other words, we have respected the "diversity" also in development of the study materials by the future teaches. So, in the evaluation of the students' works in the interdisciplinary activities, we checked how deeply they considered what they try to express with integrating what they have learned throughout that class. For that reason, our studies have mainly discussed on what kind of climate and seasonal phenomena relating to the music etc. can be used for the study materials in our interdisciplinary lesson practice and the typical examples of the students' works with the students'/our explanation/comments are presented in our papers rather than the general analysis results in somewhat statistical and objective way.

In the revised version, we will add further statements relating to the above with reconstructing mainly the Introduction, although we have already mentioned in the discussion paper.

Specific comments I add here some comments for improving the paper:

- I think the structure of the paper could be better balanced. My advice would be to structure the paper in three main sections for each of the analyzed climate concepts: (1) description of the climate concept and characteristics; (2) description of associated music / colors with these events, including a list of musical pieces and art works; (3) presentation of the activity including the followed methodology; and (4) analysis of students results: study on the use of colors, instruments and temporal patterns, agreement between students, and conclusions on how climate concepts are related to musical ones.

Thank you very much for your suggestion. As for the viewpoint in this paper, please also refer to our answer to your general comments. In addition, as for the evaluation of the students' works, we will add a little more explanation to the answer to your general comments, as follows (also relating to (3) and (4)). We paid attention to what the students felt, what they paid attention to, and how they tried to express them. Furthermore, we paid attention also to the relationship between what they want to express and how they had expressed it (e.g., combination of sounds, rhythm, etc.). It is also important to say that what we aimed in this activity is not the workmanship. We aimed to provide an opportunity for all the students, both with a lot of music experience and with little music experience, to feel familiar with the various ways of feeling and expression (diversity), through the trial of expressing what they feel and the performance/appreciation of such works.

Considering your suggestions (also considering the comments by another reviewer Louise Arnal), we will try to reconstruct the manuscript as follows (the titles of the sections and subsection are only tentative and will be finally determined in the revised version).

1 Introduction 2 Viewpoint to grasp the characteristics of the seasonal cycles in is paper *We will add a new section also considering the Armal's comments. 3 Interdisciplinary

lesson studies on climate and cultural understanding education with attention to the asymmetric seasonal march from autumn to the next spring around Japan 3.1 Asymmetric features of the seasonal march from autumn to the next spring and the relating "seasonal feeling" around Japan 3.2 A report of interdisciplinary lesson practice for the university students in teacher training course 3.2.1 Outline of the lesson practice 3.2.2 Difference of the seasonal feeling between early winter and early spring around Japan expressed in the students' works in Activities (2) and (3) 4 Interdisciplinary lesson studies with attention to the seasonal cycle and the "seasonal feeling" around Germany and Northern Europe 4.1 Remarks on the seasonal cycle and the "seasonal feeling" around Germany 4.2 Remarks on the seasonal cycle and the "seasonal feeling" around Northern Europe 4.3 A report of interdisciplinary lesson practice on the seasonal cycle and the "seasonal feeling" around Germany for the university students in teacher training course 4.3.1 Outline of the lesson practice 4.3.2 Discussion from the students' composition works in Activity (2) 4.4 A report of interdisciplinary lesson practice on the seasonal cycle and the "seasonal feeling" around Northern Europe for the university students in teacher training course 4.4.1 Outline of the lesson practice 4.4.2 Discussion from the students' composition works in Activity (2) 5 Discussions 6 Summary and conclusions

Although it seems that only small change might be made for the construction, we think that adding of the new "Section 2" would be very important. After the Amal's comments, we will mention what kind of climate/seasonal features we should pay attention to. We have already mentioned on this in the discussion paper from L399 to 410 of P13 in the section "Summary and discussions". However, now we reached to think it would be much better to move this part to the new "Section 2", with adding more detailed explanations. Then it would be also made clear that we are focusing the rather different view points on the characteristics of the seasonal cycles treated between the new sections 3 (asymmetric seasonal progression in Japan) and 4 (Germany and northern Europe). That is, as for the new section 3, we are paying attention to the asymmetric seasonal progression from autumn to the next spring around Japan, which is due to

the fact that some meteorological/climatologocal elements in the same region show sometimes rather large phase lag of their seasonal cycles among each other. On the other hand, in the discussions for the seasonal cycles in Germany and northern Europe comparing with those around Japan, we are focusing on the features found from the air temperature data that the duration of the highest stage of the seasonal mean temperature around Germany is rather longer (from the beginning of June to late August) than in Northern Europe (from late June to late July), and that the day-to-day temperature variation is also large in both regions to result in the frequent appearance of the extreme low temperature days in winter there (which is the rather different characteristics from those around Japan).

Thus we still think it better that the topics on Germany and northern Europe are not separated into the different sections (although the subsections on these regions are separated). We would also like to mention that we think it better to describe the climate/seasonal characteristics and the related seasonal feelings including some introduction of the songs or traditional seasonal events in the same subsection, without separating them into the different subsections, because they are closely related to each other and we should comprehend them as the set of "climate and music".

- In general I think the paper focuses a lot on the climate concepts but it does not describes the relevant musical concepts, e.g. which songs are linked to the climate events in each culture, which are the properties of these songs (emotion, instruments, tempo). We miss also a description of the musical instruments used in the study, and a description of other musical concepts considered (e.g. tempo and timbre). It is very important to describe this in order to fully understand the links and results of the experiments.

Thank very much for your valuable comments. At one viewpoint, we do agree with you. However, the purpose of our study is as mentioned above (as an answers to the general comments and the first specific comments) and the description in this paper needs to be constructed basically as it is, at least at the first step. As for the evaluation

of the students' works in the interdisciplinary activities, we checked "how deeply they considered what they try to express with integrating what they have learned throughout that class", as mentioned in the answer to the general comments. But in the future, if we have a chance to collaborate with you or those who are in the similar fields to you, it would be also interesting for us to research further, also including your suggestions on the analyses of the lesson practice directly.

The small percussion instruments used in our lesson practice are rather popular in the music education and could be referred to easily from the commercial base websites (not from the papers or books in the special fields) and we did not make detailed explanation on these instruments in this paper. We will add the reason for use of the small percussion instruments in the lesson practice in the text. As we mentioned above, by using such instruments, all the students, regardless of their musical experience (skills), could take part in the activity easily; they can easily touch the instruments (which were born in the human's life) and can devise various ways to play percussion instruments.

- 2.2.1: please describe better the expression of the four seasons (Itten 1961) and relate it to the results in Figure 9 commented in 2.2.2.

The "four seasons painting" is the exercise developed by Itten, when he was teaching in the preliminary course at the Bauhaus in Germany from 1919 to 1922. Originally, this method is to express the differences between the seasons with combination of the various colors created by mixing paints. In our activity, the colored papers were used in order to make the activity much more easier for the students. In the revised manuscript, such explanation will be added.

- I miss a more rigorous analysis of the results of the three experiments, as Figures 11-19 and 20 only present and comments few examples. We would need to understand the results for all students following the same methodology, and see if we can get some conclusions on the general findings and individual differences in the way they related music, climate and emotions. It would then be interesting to analyze the agreement

between the students, and if they select the same instruments and temporal patterns in the same way.

Thank you very much for your valuable comments. In this paper, however, as we answered to the general comments, our studies have mainly discussed on what kind of climate and seasonal phenomena relating to the music etc. could be used for the study materials in our interdisciplinary lesson practice and the typical examples of the students' works with the students'/our explanation/comments are presented in our papers rather than the general analysis results of the lesson practices in somewhat statistical and objective way. We are mainly aware of "how deeply they considered what they try to express with integrating what they have learned throughout that class", rather than the workmanship of the students (although the workmanship and the interest in the activity were rather different among the students). Thus we are afraid if you might still miss the results on the general analyses for all students, etc., but it would be grateful to you to understand also our aim of this study. In the future works, however, we also feel it is necessary to investigate how to advise to the various levels of students differently, etc. (e.g., students with high music interest and skill vs. those with low ones) through our further activities.

I advise the authors to consider and relate to the large corpus of literature relating music to emotions that one would need to cite here, such as: [1] Juslin, P. (2019). Musical emotions explained. Oxford University Press. [2] Meyer, L. (1956). Emotion and the Meaning of Music. Oxford University Press. [3] Eerola, T., and Vuoskoski, J.K. (2013). A review of Music and Emotion studies: approaches, emotion models, and stimuli. Music Perception: An Interdisciplinary Journal, 30, 307-340. [4] Balkwill, L.L. and Thompson, W.F. (1999). A cross-cultural investigation of the perception of emotion in music: psychophysical and cultural cues. Music Perception,17, 43-64.

Thank you very much for information on the books and papers about music analysis. Meyer (1956) is one of the foundations of music education, and one of the present authors has read that book. In this paper, we will also refer to some of them for presenting

the aim and construction of this paper.

Technical corrections - I suggest the authors to add some complementary material to the paper which that can be useful for future educational and research activities on the area. I suggest to include a list of representative songs from Japan and Germany representative of each climate season

Although we have not decided what representative songs in Japan and Germany (how many, what seasons, etc.) will be listed up in the revised version, we are considering to show the information on the titles and composers of several songs for the specified season in Japan and Germany, maybe, as follows. * Early winter vs. early spring in Japan (mainly school songs) *Spring in various stages in Japan vs. spring (mainly for May) in Germany (As for Japan, mainly school songs or children songs. As for Germany, children songs and art songs)

- Figures 19 and 20 are difficult to ready. I would suggest to add a description in the text.

Although the explanation of the works given by the students for Figs. 19 and 20 is described in the text, we will consider how the further explanation can be added in the text.

As a conclusion, I think the paper topic is very interesting and there is a lot of work behind it, but I think it would benefit a lot from a change in the structure and a more detailed description of the results, so that the paper is self-contained for publication.

Thank you very much for your valuable comments.

Please also note the supplement to this comment:
https://gc.copernicus.org/preprints/gc-2020-18/gc-2020-18-AC4-supplement.pdf